# NEMO-Bohai 1.0: a high-resolution ocean and sea ice modelling system for the Bohai Sea, China

Yu Yan[1,2,3], Wei Gu[2], Andrea M. U. Gierisch[4], Yingjun Xu[2], Petteri Uotila[3]

[1]School of Ocean Sciences, China University of Geosciences, Beijing 100083, China

[2]Academy of Disaster Reduction and Emergency Management, Faculty of Geographical Science, Beijing Normal University, Beijing 100875, China

[3]Institute for Atmospheric and Earth System Research (INAR), Faculty of Science, University of Helsinki, 00014 Helsinki, Finland

[4]Danish Meteorological Institute, DK-2100 Copenhagen, Denmark

*Correspondence to:* Yu Yan (yuyan@cugb.edu.cn) and Petteri Uotila (petteri.uotila@helsinki.fi)

**Abstract.** Severe ice conditions in the Bohai Sea could cause serious harm to maritime traffic, offshore oil exploitation, aquaculture, and other economic activities in the surrounding regions. In addition to providing sea ice forecasts for disaster prevention and risk mitigation, sea ice numerical models could help explain the sea ice variability within the context of climate change in marine ecosystems, such as

spotted seals, which are the only ice-dependent animal that breeds in Chinese waters. Here, we developed NEMO-Bohai, an ocean-ice coupled model based on the Nucleus for European Modelling of the Ocean (NEMO) model version 4.0 and Sea Ice modelling Integrated Initiative (SI$^3$) (NEMO4.0-SI$^3$) for the Bohai Sea. This study will present the scientific design and technical choices of the parameterizations for the NEMO-Bohai model. The model was calibrated and evaluated with in-situ and satellite

observations of the ocean and sea ice. The model simulations agree with the observations with respect to sea surface height (SSH), temperature (SST), salinity (SSS), currents, and temperature and salinity stratification. The seasonal variation of the sea ice area is well simulated by the model compared to the satellite remote sensing data for the period of 1996-2017. Overall agreement is found for the occurrence dates of the annual maximum sea ice area. The simulated sea ice thickness and volume are in general

agreement with the observations with slight over-estimations. NEMO-Bohai can simulate seasonal sea ice evolution and long-term interannual variations. Hence, Nemo-Bohai is a valuable tool for long-term ocean and ice simulations as well as climate change studies.

## 1. Introduction

The Bohai Sea is the southernmost seasonal frozen sea in the Northern Hemisphere (Yan et al., 2017). The formation of sea ice in the Bohai Sea mainly depends on the geographical environment and hydrometeorological characteristics (Ding, 1999). Specifically, the Bohai Sea is located in the continental shelf area, and the average water depth is only 18 m (Su and Wang, 2012), which indicates low oceanic heat content in winter. Following a northern continental climate, the Bohai Sea is affected by the cold Siberian air every winter, which causes the sea surface temperature of the Bohai Sea to be significantly lower than that at the same latitude (Zhang et al., 2016; Donlon et al., 2012). In addition, the sea surface salinity of the Bohai Sea is about 30 PSU, which is the lowest in the entire coastal waters in China (Yan et al., 2020). It means that the Bohai seawater freezes before reaching the maximum density. Therefore, it even more easily convects and loses heat just before freezing. The sea ice season is generally from December to March. The sea ice in the Bohai Sea also exhibits significant interannual variability, which could cover half of the sea area during severe winters (Su and Wang, 2012).

With four of China's top ten ports (Tianjin, Tangshan, Dalian, Yingkou), the Bohai Sea is an important economic zone in China (Fu et al., 2017). Under severe sea ice conditions, such as during the 2009/2010 winter, 296 ports in this region were frozen. In particular, the 42 km artificial waterway of Huanghua Port was covered by sea ice; thus, the maritime traffic was severely restricted, and 7157 fishing vessels were damaged (Gu et al., 2013). About $2 \times 10^5$ hectares offshore aquatic farms were covered by sea ice, and the aquaculture industry losses reached ~6 billion yuan (~878 million US dollars) (Zhang et al., 2013). In addition, ice floes drift on the Bohai Sea surface under the drive of winds, tides, and ocean currents. When the ice floes meet offshore structures (oil and gas platforms, navigation lights, and lighthouses) or ships, convergent ice motion seriously threatens marine transportation and oil and gas exploration (Leppäranta, 2011; Li et al., 2011; Yan et al., 2019). Thus, sea ice monitoring and forecasting are essential for assessing sea-ice-hazard risk and preventing ice disasters.

In addition, spotted seals (*Phoca largha*) are listed as the least concern species by the International Union for the Conservation of Nature Red List of Threatened Species and as a Grade II state protection species by China's Wild Animal Protection Law (1988), which are distributed in the Northwest Pacific Ocean with the Liaodong Bay of the Bohai Sea as the southernmost geographic breeding site. At the end of October each year, the seals traveled from the Yellow Sea to Liaodong Bay. However, the population

of the spotted seal has experienced several drastic declines from ~2300 to ~1000 individuals over the past three decades due to hunting and environmental pollution (Yan et al., 2018; Yang et al., 2017). They den in coastal areas and on offshore drifting ice. When the sea ice melts, they gradually leave Liaodong Bay and embark on a long journey back to the Pacific Ocean. Therefore, it is crucial to study long-term changes of Bohai Sea ice to study its impacts on spotted seals.

Most research has focused on the polar region, however, sea ice in mid-high latitude is also sensitive to climate change (Bai et al., 2011; Gong et al., 2007; Yan et al., 2017). Knowledge of regional sea ice variability is vital to studying how climate change will affect regional basins. The traditional view on the Bohai Sea ice is concentrated on extracting sea ice data from satellite imagery (Karvonen et al., 2017; Shi and Wang, 2012; Su et al., 2019). As the use of remote sensing data to retrieve sea ice data has limitations in providing a continuous record for a long time series, numerical modelling is an effective tool to understand the sea ice development from freezing to thawing (Hordoir et al., 2019; Pemberton et al., 2017).

Sea ice models can reasonably simulate sea ice conditions, which can primarily supplement remote Earth observations. To our knowledge, there are only very few studies on developing regional coupled ocean-ice models for the Bohai Sea, and most of them have focused on short-duration simulations, such as one-week or one-year case studies. Wang et al. (1984) reported the first sea ice dynamic-thermodynamic model for the Bohai Sea, which simulated the sea ice freezing-melting cycle at a resolution of 20 km×20 km driven by monthly climatic forcing. Wu (1991) and Wu et al. (1997) tried to simulate sea ice evolution based on a dynamic-thermodynamic model with consideration of the ice thickness distribution function consisting of three types (level ice, drift ice, and open water). Bai and Wu (1998) combined the sea ice dynamic-thermodynamic model, atmospheric boundary layer model, and tide model at a resolution of 8.64 km×11.11 km. Liu et al. (2003) applied a high-resolution atmosphere model to reproduce sea ice conditions in 2002 by a sea ice model initialized by high-resolution satellite data from HY-1 and MODIS. Su et al. (2004) simulated the winters of 1998 and 2000 with the coupled Princeton Ocean Model (POM) and viscous-plastic dynamic-thermodynamic sea ice model. Tang et al. (2010) simulate the sea ice freezing-melting process in 2003 with the coupled POM and a sea ice model considering the thickness distribution function. Zhang and Zhang (2013) used the finite-volume, primitive equation community ocean model to simulate sea ice in 2003. Li et al. (2021) simulated sea ice

variations in the severe winter of 2009–2010 in the Bohai Sea based on version 3.6 of the Nucleus for European Modelling of the Ocean (NEMO). To the best of our knowledge, there has been no long-term simulation study on Bohai Sea ice so far aiming to understand the interannual or interdecadal variability.

NEMO is a state-of-the-art numerical modelling framework designed for research activities and forecasting services in the ocean and climate sciences (Madec et al., 2016). The NEMO code, including its sea ice component, enables the investigation of ocean-ice dynamics/thermodynamics and their interactions with the atmosphere. NEMO offers a wide range of applications from short-term forecasts and climate projections (Drouard and Cassou, 2019; Obermann-Hellhund et al., 2018; Uotila et al., 2017;

Voldoire et al., 2013) to process studies (Courtois et al., 2017; Declerck et al., 2016; Feucher et al., 2019). While global models offer a poor representation of regional ocean processes, regional models have been developed based on the framework of NEMO in recent years for various seas, e.g., the Atlantic marginal basins (Graham et al., 2018; O'Dea et al., 2017), the Baltic and North seas (Hordoir et al., 2019; Pemberton et al., 2017), the Northwestern Mediterranean Sea (Declerck et al., 2016), the South Indian

Ocean (Schwarzkopf et al., 2019), the South China Sea (Thompson et al., 2019), and the Black Sea (Gunduz et al., 2020). However, a NEMO-based regional model for the Bohai Sea has not been attempted for long-term climate studies until now.

Long-term sea ice simulations could help improve our understanding of thermodynamic and dynamic sea ice processes in the Bohai Sea, crucial for sea ice disaster prevention, spotted seal habitat

studies, and regional climate change studies. The paper aims to report the development of NEMO-Bohai and assess its performance. The paper is organized as follows: Sec. 2 introduces the model and observation dataset. Model comparison and validation are carried out in Sec. 3. The analysis of sea ice variation based on the 22-year hindcast simulations of NEMO-Bohai is presented in Sec. 4. Discussion and a summary are provided in Sec. 5.

**2. Model setup: NEMO-Bohai**

NEMO-Bohai is localized to the Bohai Sea based on the NEMO ocean engine (Madec et al., 2016). We apply the NEMO 4.0 beta and Sea Ice Integrated Initiative ($SI^3$) model in a regional configuration covering the Bohai Sea. The regional configuration, NEMO-Bohai, is nested into the global configuration using one-way lateral boundary conditions with the flow relaxation scheme algorithm (Engedahl, 1995).

The global 1/4° ocean-ice model (ORCA025) maintained by the DRAKKAR group is performed. A detailed description of ORCA025 is provided by Barnier et al. (2006). In our simulation, forcing fields are provided from the Japanese 55-year Reanalysis (Harada et al., 2016; Kobayashi et al., 2015), covering the 55 years from 1958 with 0.5°×0.5° spatial resolution. A monthly climatological runoff flux with Antarctic ice shelves and iceberg meltwater flux (Dai et al., 2009; Depoorter et al., 2013) is used. To limit the atmospheric forcing biases from propagating into the upper ocean and avoid salinity and temperature drift, which impacts the overturning circulation, the sea surface temperature (SST) and salinity (SSS) are restored monthly based on the World Ocean Atlas-2013 (Locarnini et al., 2013; Zweng et al., 2013).

The global simulation is spun up in January 1987 with the rest state and ended in 2017. The time step is set to 900 s. The ORCA025 configuration is performed using 240 CPU cores on a Cray XC40 system on the Sisu supercomputer and requires ~6000 CPU hours per simulated year. The model output frequency is set to every five days from 1987 to 1994 for the test run and increases to every day for the period of 1995−2017. The size of the output files per simulated year is approximately 2 TB. The open boundary for NEMO-Bohai is forced by barotropic and baroclinic modes. The sea surface heights (SSHs), temperatures, salinity levels, and velocities are relaxed to outer-model values over a 1-point zone at the unstructured open boundary of the model domain. In addition, the use of the XIOS server makes NEMO-Bohai more user-friendly.

**2.1 NEMO-Bohai: The ocean model**

The NEMO-Bohai domain consists of one central zone and three bays (see Fig. 1). The area covers 117°–122° E and 37°–41° N. The horizontal resolution is 0.025° in orthogonal curvilinear coordinates, which is equivalent to a resolution of approximately 2 km (204 × 244 = 49 776 grid points). The bathymetry is interpolating into the target grids according to the ETOPO1 1 arc-minute dataset. We further manually remove the isolated land cells with a number less than 4. The vertical discretization scheme adopts the $z*$ formulation. There are ten vertical levels, and the top layer is set to 1 m. The partial step is applied to the bottom topography. The time step of the ocean model is set to 90 s, while it calls the sea ice model every three time steps. The split-explicit time stepping method (`ln_dynspg_ts = .true.`) is applied to compute barotropic/baroclinic mode and their interactions.

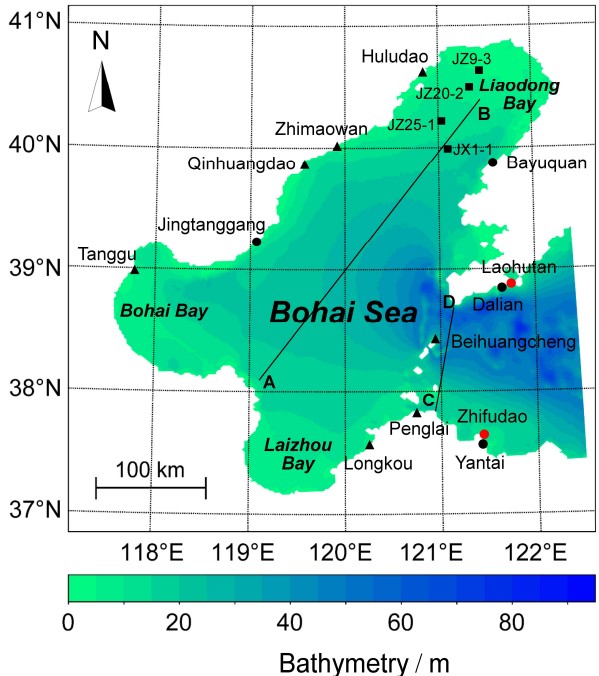

**Figure 1: The location and bathymetry of the Bohai Sea. Black dots denote the coastal tide gauge stations; red dots and black triangles indicate station locations with sea surface temperature and sea surface salinity observations, respectively; black squares denote the oil platforms where the sea ice thickness observations were conducted. Line AB represents the profile section along which temperature and salinity were presented (figures 6 and 7, respectively), while line CD indicates the Bohai Strait. The straight line in the east denotes an open boundary for the regional ocean model.**

The initial conditions of the numerical simulation in the Bohai Sea for July 1, 1995, including SST and SSS, are obtained by interpolating the temperature and salinity fields from the global ocean simulations with the ORCA025 configuration, as shown in Fig. 2. The regional model is forced with lateral ocean boundary conditions, tidal forcing, atmospheric forcing, and river runoff during the study period. Two kinds of boundary conditions are used for the setting of lateral open ocean boundary. Flow relaxation scheme is applied to baroclinic velocities and tracers (Engedahl, 1995), while Flather boundary condition (Flather, 1976) is used for barotropic dynamics, such as SSH and barotropic velocities (`u2d`, `v2d`). NEMO-Bohai has one open boundary located 100 km away from the Bohai Strait, and we set the relaxation zone width to 1 (`nn_rimwidth = 1`). The lateral open ocean boundary conditions for 1995 to 2017 are also taken from a series of global ocean simulations with the ORCA025 hindcast. The Oregon State University Tidal Inversion Model is used for the barotropic mode (Egbert and Erofeeva, 2002), including 11 tidal harmonics (M2, S2, N2, K2, K1, O1, P1, Q1, M4, MS4, MN4). We choose the TPXO9.1 global tide model, which is the latest version with a 0.25°×0.25° resolution.

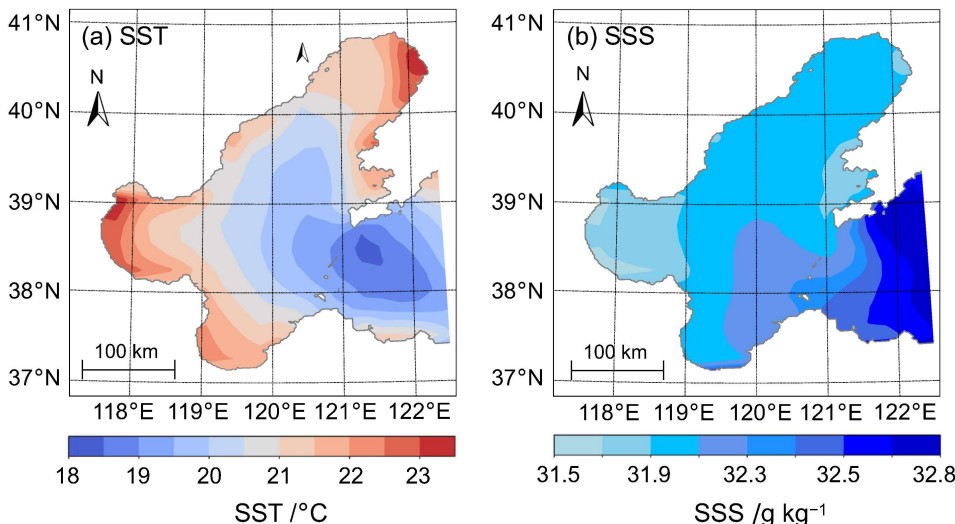

**Figure 2: Examples of (a) initial sea surface temperature (SST) field and (b) initial sea surface salinity (SSS) field interpolated from the global ORCA025 model simulation on July 1, 1995.**

The NEMO-Bohai atmospheric forcing is derived from the ERA5 dataset, the ECMWF's latest reanalysis product covering the period from 1979 to the present, which has replaced the widely used ERA-Interim dataset. The data cover the Earth with a horizontal spatial resolution of 30 km and represent the atmosphere using 137 levels from the surface up to a height equaling 0.01 hPa. The forcing files consist of 1 hourly instantaneous field of the 2 m air temperature, 10 m wind speed, downward short/long

wave radiation, sea-level pressure, specific humidity, and precipitation. The equivalent inverse barometer SSH is calculated from the atmospheric pressure (`ln_apr_dyn = .true.`) with the referenced pressure of 101,000 N m$^{-2}$. In addition, the river runoff, which provides a significant volume of freshwater to the Bohai Sea and influences stratification, circulation, and primary productivity, is considered in the model. The daily runoff volume data for the rivers flowing into the Bohai Sea, such as

the Yellow River, Liao River, and Hai River, are acquired from JRA-55 from 1995 to 2018. The runoff forcing is considered as freshwater with a salinity of 0 PSU and the same temperature as the ocean surface.

     The bottom roughness impacts the dynamics of the tide, ocean circulation, and storm surges in the Bohai Sea. A constant bottom roughness (`rn_z0`) is used to calculate the drag coefficient for all bottom grids. To better reflect changes in the tide and circulation, we tune the bottom roughness accordingly.

Regarding the configuration in the Gulf of Finland (`rn_z0 = 5.0×10⁻³ m`) (Westerlund, 2019), we set `rn_z0` to $5.0\times10^{-1}$, $5.0\times10^{-2}$, $5.0\times10^{-3}$, $5.0\times10^{-4}$, $5.0\times10^{-5}$, and $5.0\times10^{-8}$ while keeping all the other variables consistent. Results show that reducing the bottom friction can increase the phase difference and amplitude of the tide. In NEMO-Bohai, the smaller the roughness value is, the more accurate the tide is.

However, it also needs to be combined with the actual situation at the bottom of the Bohai Sea. With reference to Zhang and Zhang (2013), the bottom roughness had a magnitude of $1 \times 10^{-3}$ m. Therefore, the bottom roughness in NEMO-Bohai is determined to be $5.0 \times 10^{-3}$ m. The bottom drag coefficient (`rn_Cd0`) is set to the default value after a set of experiments. The turbulent kinetic energy (TKE) closure scheme is used for vertical mixing (Blanke and Delecluse, 1993) with the default values of vertical eddy viscosity and diffusivity coefficients, and the flux-corrected transport scheme (Zalesak, 1979) is responsible for the tracer advection. Some critical parameters of the ocean part of NEMO-Bohai are listed in Table 1.

**Table 1: Key physical parameters in the ocean namelist (namelist_oce) in NEMO-Bohai.**

| Parameter | Namelist | Setting | Parameter | Namelist | Setting |
|---|---|---|---|---|---|
| Time step | rn_rdt | 90 s | Activate tides | ln_tide | true |
| Frequency of surface module call | nn_fsbc | 3 | Hydrostatic pressure gradient option | ln_hpg_sco | true |
| Relaxation zone width | nn_rimwidth | 1 | Ocean equation of seawater | ln_eos80 | true |
| Bottom roughness | rn_z0 | $5.0 \times 10^{-3}$ m | Penetrative solar radiation | ln_qsr_rgb | true |
| Top drag coefficient | rn_Cd0 | $1.0 \times 10^{-3}$ | Advection scheme for tracer | ln_traadv_fct | true |
| Bottom drag coefficient | rn_Cd0 | $1.0 \times 10^{-3}$ | Standard operator of lateral diffusion scheme for tracers | ln_traldf_iso | true |
| Lateral momentum boundary condition | rn_shlat | 0 | Laplacian operator of lateral diffusion on momentum | ln_dynldf_lap | true |
| Vertical eddy viscosity | rn_avm0 | $1.2 \times 10^{-4}$ m$^2$ s$^{-1}$ | Surface pressure gradient | ln_dynspg_ts | true |
| Vertical eddy diffusivity | rn_avt0 | $1.2 \times 10^{-5}$ m$^2$ s$^{-1}$ | Vertical physics | ln_zdftke | true |
| Ocean initialization | ln_tsd_init | true | Top/bottom drag coefficient | ln_loglayer | true |

**2.2 NEMO-Bohai: The sea ice model**

SI$^3$ is recommended by the Sea Ice Working Group to reduce duplication and better use development resources (Aksenov et al., 2019). SI$^3$ merges the capabilities of three sea ice models previously used in NEMO (CICE, GELATO, and LIM). SI$^3$ is a horizontal dynamic and vertical thermodynamic sea ice model. The thermodynamics and dynamics are separated and rely upon different frameworks and sets of hypotheses: thermodynamics uses the ice thickness distribution and the mushy-layer frameworks, whereas dynamics assumes continuum mechanics, e.g., Leppäranta (2011). Thermodynamics and dynamics interact in two ways: first, advection impacts the ice state variables; second, the horizontal momentum equation depends on, among other things, the ice state. The modified elastic-viscous-plastic rheology is used for sea ice dynamics (Bouillon et al., 2013; Kimmritz et al., 2017). Bohai Sea ice is

relatively thin and has low salinity, different from polar ice (Gu et al., 2013). Therefore, the parameters in NEMO-Bohai are required to be modified accordingly.

In NEMO-Bohai, we selected and adjusted a series of sea ice model parameters. We increase the number of ice categories (jpl) and the number of ice layers (nlay_i) since Massonnet et al. (2019) suggested that increasing the resolution of the ice thickness distribution results in small but non-negligible variations in the ice extent and volume. The ice thickness is discretized into ten categories (0.0, 0.0476, 0.0976, 0.150, 0.205, 0.263, 0.324, 0.388, 0.455, 0.526, 0.601 m), and the thermodynamic

calculations used five ice layers (nlay_i = 5). The minimum ice thickness used in remapping is set at the minimum value (rn_himin = 0.01 m) to capture new thin ice formations better. The thickness of new ice formed in open waters (rn_hinew) is set to 0.02 m as the value must be larger than rn_hnewice. According to the test run of NEMO-Bohai, the sea ice thickness is generally overestimated. Compared to the referenced value of 2.0 m, we reset the average ice thickness

(rn_himean) as 0.20 m according to the observations (Li et al., 2004; Yuan et al., 2012).

The ice initialization is activated (ln_iceini = .true.), and the initial ice salinity (rn_smi_ini_n) is set to 7 PSU, and the initial real snow thickness (rn_hts_ini_n) is set to 0.1 m according to in-situ observations (Bai and Wu, 1998; Gu et al., 2014). For the snow and ice albedos, we set them to lower values compared to the referenced ones based on the in-situ observations (Zheng et

al., 2017). The ice strength parameter (rn_pstar) is defined as $2.75 \times 10^4$ N m$^{-2}$ based on previous in-situ studies (Li et al., 2011). A ridging scheme is considered in SI$^3$. Thus, the parameter rn_hstar, which adjusts the maximum thickness of ridged ice, is reduced from 100.0 m to 10.0 m following the NEMO-NORDIC 1.0 configuration (Pemberton et al., 2017) to reflect the general thinner Bohai Sea ice. In addition, the free-slip lateral boundary condition is chosen for sea ice dynamics (rn_ishlat = 0.),

which is synchronized with the ocean model, and a landfast parameterization is set (ln_landfast_L16 = .true.) because there exists landfast ice in the eastern Liaodong Bay.

## 3. Model comparison and validation

In this section, we analyze how well NEMO-Bohai reproduces ocean properties (SSH, SST, SSS, temperature and salinity stratification, currents and volume exchanges with the Yellow Sea) and sea ice properties (area, thickness, and volume). To evaluate the model's performance, we compare our model results to multiple sources in-situ and satellite observations.

### 3.1 Observational data

To evaluate the SSH, the modeled tide amplitude and phase were compared to the tide tables with hourly intervals of Yantai (37.550° N, 121.383° E), Dalian (38.867° N, 121.683° E), and Jingtanggang (39.217° N, 119.017° E) from National Marine Data and Information Service during October 1 to 31, 2012. In addition, we use observed in-situ data from the National Marine Environmental Monitoring Center. The sampling station is at Bayuquan (39.804° N, 121.456° E), and the sampling period is from January 1 to 31, 2013. For the SST evaluation, we use data from the National Marine Science Data Center. The coastal stations are located at Laohutan in Dalian (38.90° N, 121.70° E) and Zhifudao in Yantai (37.60° N, 121.40° E). The observed records cover 2000–2001, 2010 (July–December), and 2011–2015. For the sea surface salinity, the observed in-situ data for eight ocean stations (Wentuozi, Huludao, Zhimaowan, Qinhuangdao, Tanggu, Longkou, Penglai, Beihuangcheng) in the Bohai Sea are from Yuan et al. (2015). Specifically, the observation period in Huludao, Zhimaowan, Qinhuangdao, Tanggu, Longkou, and Beihuangcheng is from 1960 to 2013, while the period in Wentuozi and Penglai is 1987-2001 and 2012-2013, respectively. The climatological vertical profiles of temperature and salinity are from the marine atlas of the Bohai Sea based on the observed data from the 1950s to 1990s (Chen, 1992).

The sea ice model is evaluated against a series of observational datasets. A satellite-derived dataset covering the winters from 1988 to 2017 was retrieved. Sea ice area was extracted from two datasets; the first based on the zonal threshold method for Advanced Very High Resolution Radiometer (AVHRR) during 1988-2000; and the second based on an object-based feature extraction method for Moderate Resolution Imaging Spectroradiometer (MODIS) from 2001 to 2017. A detailed description can be found in Yan et al. (2017). The shapefiles derived by the abovementioned methods are further modified by visual interpretation to build a more accurate sea ice area dataset. For the sea ice thickness evaluation,

we use the in-situ observations from four offshore oil platforms (see Fig. 1) on January 2, 6, 9, 16, 26, February 2, 9, 12, 16 during 2013 (Zeng et al., 2016; Karvonen et al., 2017).

### 3.2 Ocean

### 3.2.1 Sea surface height

Reliable precision of the tidal model is pre-required for subsequent simulations by NEMO-Bohai. The tidal model is validated by in-situ hourly tidal observations at Bayuquan and tide table data at Yantai, Dalian, and Jingtanggang. Fig. 3 provides a robust description of the comparison of SSH time evolution at the four stations. The tidal range in the Bohai Strait (Yantai and Dalian) is larger than that at other tide gauging stations. It is clear that the modeled water elevation at Bayuquan and Jingtanggang stations agrees less with the tide table data than at the other two stations, as shown in Table 2. Nevertheless, the model reproduced well the semidiurnal M2 tidal cycle with exact phase and amplitude at four stations with a mean correlation of 0.92, mean absolute error of 0.17 m, and root-mean-square error of 0.22 m. The model reproduces SSH standard deviation well, and the difference is within 6 cm in Yantai, Dalian, and Jingtanggang (Table 2); the largest deviation at Bayuquan is approximately 11 cm. Simulated SSH depends on the model's bathymetry, open boundary forcing, and freshwater flux (Kärnä et al., 2021). The errors are partly due to the deviation of positions between the tidal gauges and the model grid points, as the model has a horizontal grid size of ~2 km. Also, errors can be partly due to the inaccurate bottom topography, especially in the shallow water depth zone. As shown in Fig. 6, there is a clear topographic error between the model and the atlas for the profile section AB. In general, the reasonable accuracy of the hydrodynamic model can partially assure rather realistic simulations of the sea ice growth and melting, as described in section 3.3.

**Table 2: Sea surface height representation, in terms of standard deviation (SD, meters), correlation, mean absolute error (MAE, meters), and root-mean-square error (RMSE, meters), made by NEMO-Bohai for four Bohai Sea stations.**

| Station | SD Observation (meters) | SD NEMO-Bohai (meters) | Correlation | MAE (meters) | RMSE (meters) |
|---|---|---|---|---|---|
| Yantai | 0.65 | 0.59 | 0.96 | 0.14 | 0.18 |
| Dalian | 0.81 | 0.85 | 0.96 | 0.18 | 0.23 |
| Jingtanggang | 0.36 | 0.37 | 0.84 | 0.17 | 0.21 |
| Bayuquan | 0.61 | 0.50 | 0.92 | 0.20 | 0.25 |

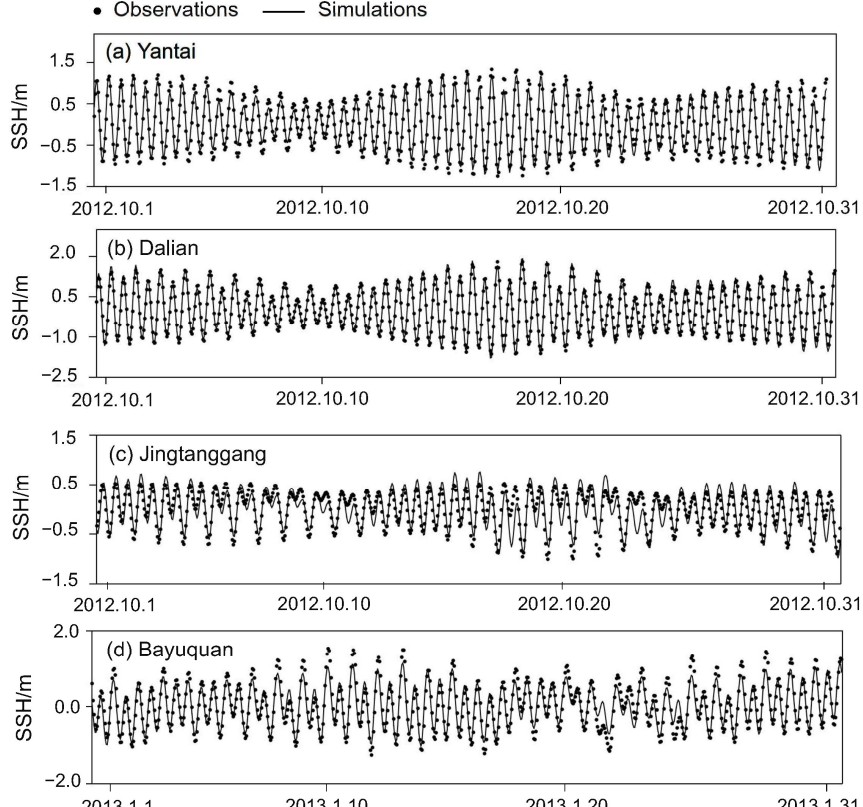

**Figure 3: Comparison of sea surface height (SSH) between the observations and NEMO-Bohai simulations at (a) Yantai, (b) Dalian, (c) Jingtanggang, and (d) Bayuquan stations in the Bohai Sea.**

### 3.2.2 Sea surface temperature and salinity

As shown in Fig. 4, the modeled SST followed well the seasonal cycle, and the mean absolute error

was typically less than 1 °C at Laohutan and Zhifudao stations during the observed period. The modeled variations of SST are smoother than observed, and they particularly fail to capture the high fluctuations of daily variations in summer/autumn at both stations. It is also noticeable that the simulated SST at Zhifudao is generally colder in winter and warmer in summer than the observations.

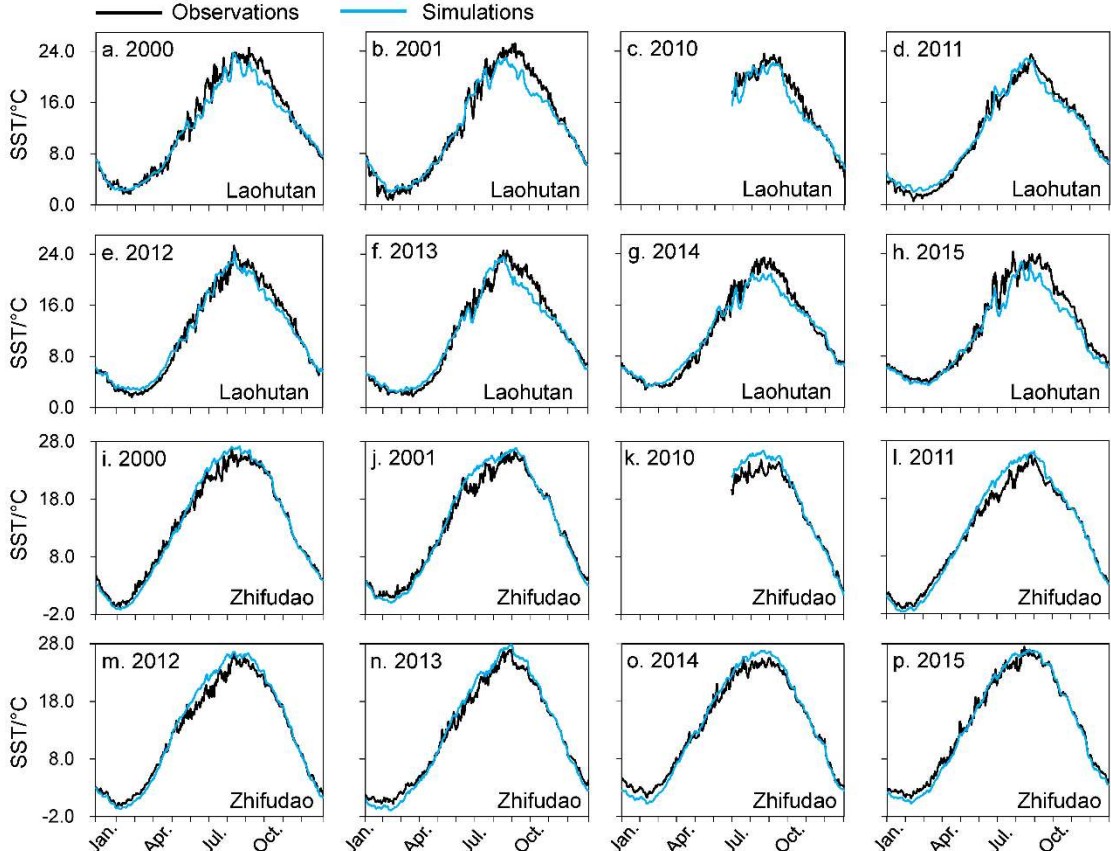

**Figure 4: The simulated sea surface temperature (SST) compared with in-situ observations at Laohutan, Dalian (a-h), Zhifudao, and Yantai (i-p) for the period of 2000-2001 and 2010-2015, respectively.**

Table 3 displays the comparisons of multi-year average sea surface salinities at eight ocean stations in the Bohai Sea between simulations from NEMO-Bohai and observations reported by Yuan et al. (2015). Generally speaking, NEMO-Bohai is able to capture the main variations of the sea surface salinity in the Bohai Sea. Values at six ocean stations agree with observations with the relative deviation less than 5%, while the modeled values at Huludao and Tanggu are less salty than observed. This demonstrates that the model faces more significant challenges in the low salinity areas. Primarily, the freshwater river runoff leads to lower salinity in the coastal regions. In Nemo-Bohai, the river runoff is based on climatological estimates without interannual variability. The river salinity is assumed to be 0 PSU, and the river temperature is set to the same value as the SST at the closest grid point. All these reasons might cause underestimations. In addition, the biases of river runoff and shallow water depth (generally < 3 m) also need to be taken into considerations.

**Table 3: Comparison of multi-year average sea surface salinity between the observations and simulations.**

| Station | longitude (°) | Latitude (°) | Observations (PSU) | Simulations[*] (PSU) | Relative deviation |
|---------|---------------|--------------|--------------------|----------------------|--------------------|
| Wentuozi | 121.41 | 39.72 | 30.5 | 30.1 | –1.3% |
| Huludao | 120.81 | 40.56 | 29.6 | 23.2 | –21.6% |
| Zhimaowan | 119.92 | 40.02 | 30.8 | 31.0 | +0.6% |
| Qinhuangdao | 119.62 | 39.92 | 30.4 | 31.0 | +0.6% |
| Tanggu | 117.78 | 38.98 | 29.3 | 25.4 | –13.3% |
| Longkou | 120.32 | 37.65 | 28.2 | 28.0 | –0.7% |
| Penglai | 120.73 | 37.82 | 30.2 | 30.4 | +0.6% |
| Beihuangcheng | 120.92 | 38.40 | 31.5 | 32.8 | +4.1% |

[*]The observed sea surface salinity data is referenced from Yuan et al. (2015).

### 3.2.3 Current

The simulated monthly mean current velocities at the surface and 16 m depth in February and August are shown in Fig. 5. The monthly mean current velocities are calculated based on hourly model output during August 2012 and February 2013. The figure shows that both the sea surface and 16 m depth currents are usually less than 0.4 m s$^{-1}$. Due to the blocking effects of the bays, the currents are weak at the head of the three bays, which is consistent with the observations by Chen et al. (1992). The maximum current velocity zone is located in the northern Bohai Strait, in a good agreement with the model simulation result of Ji et al. (2019). The inflow and outflow occur in the northern and southern parts of the Bohai Strait in both seasons, respectively, which is consistent with the observations (Zhang et al., 2018). Specifically, the strongest modeled inflow from the Yellow Sea through the Bohai Strait occurs in a narrow channel in its northern part, namely the Laotieshan Channel, which agrees with the observations of Wan et al. (2015). Also, Lin et al. (2011) suggested that persistent winds drive a cyclonic coastal current in the northern Yellow Sea, and one branch of the current enters the Bohai Sea at the northern Bohai Strait, which transports warm and saline water from the Yellow Sea.

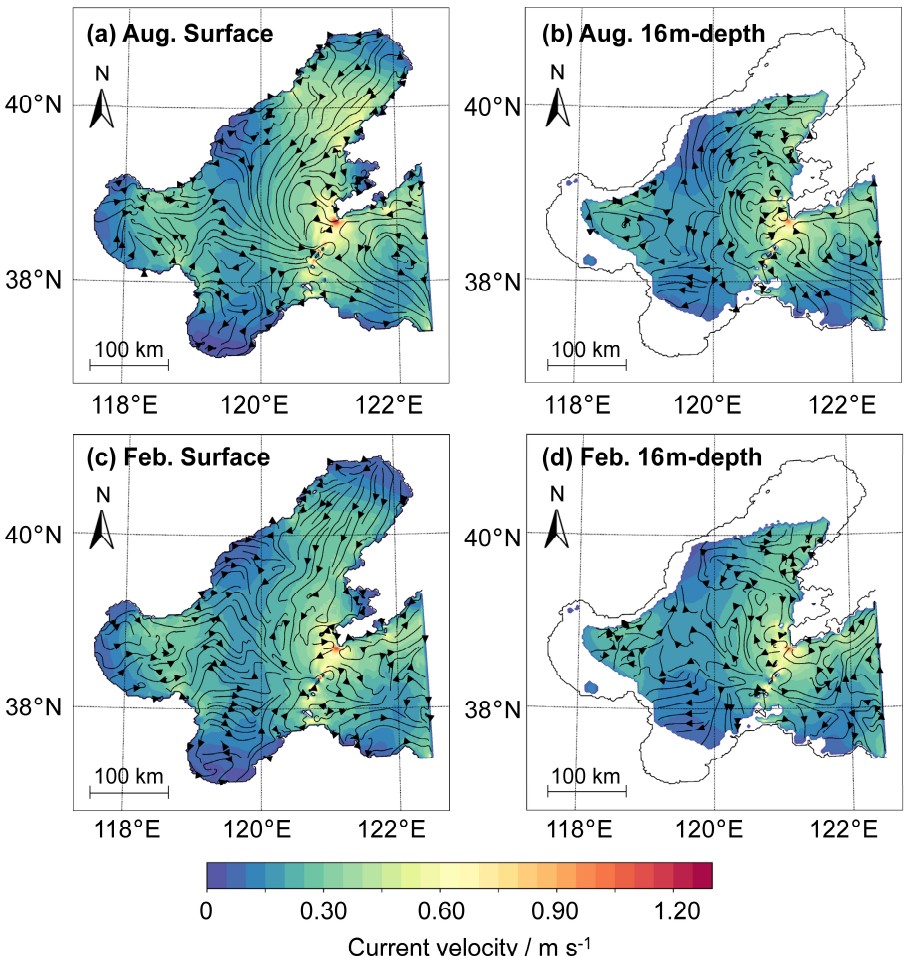

**Figure 5: Simulated monthly mean current velocities at surface and 16 m depth in August 2012 and February 2013. The monthly mean current velocities are calculated based on the outputs with hourly intervals. The black lines and arrows represent the streamlines and directions of the current vector field, respectively. The filled contours denote the current speed in m s⁻¹.**

The monthly mean water volume exchange at the Bohai Strait (see Fig.1) based on hourly model simulations during August 2012 and February 2013 are also calculated to evaluate the model's performance. The Bohai Sea water exchange with the Yellow Sea is weak due to its half-closed shape and a relatively independent circulation system. The model results show that the inflow from the Yellow Sea to the Bohai Sea in August reaches $6.9 \times 10^4$ m³ s⁻¹, almost double than that in February ($3.5 \times 10^4$ m³ s⁻¹), which lies in the range of $5 \times 10^3$ to $8 \times 10^4$ m³ s⁻¹ indicated by Bian et al. (2016). Our results show that the outflow exists in both months (August: $8.3 \times 10^4$ m³ s⁻¹; February: $6.5 \times 10^4$ m³ s⁻¹), with a larger amount than the inflow. The net flow appears to be outflow both in winter and summer, which is consistent with other model simulation results (Lin et al., 2002; Ji et al., 2019).

### 3.2.4 Vertical profile

NEMO-Bohai and observed water temperature and salinity profiles along the transect AB (see Fig.1) are shown in Fig. 6 and Fig. 7, respectively. Observations are from the atlas by Chen (1992), which is based on data from the 1950s to 1990s. The temporally closest 5-year period from 1995 to 2000 of NEMO-Bohai simulations was selected for model-observation comparisons. Common features are found both in the model and observations. The Bohai Sea waters are vertically well-mixed in autumn and winter,

and they have a remarkable homogeneous vertical distribution for both temperature and salinity. In spring and summer, thermal stratification occurs with a significant cold-water core at depth, eventually eroded in autumn. As apparent in Fig. 6 and Fig. 7, the stratification in shallow coastal waters is generally homogeneous. Similar features were reported by Wang et al. (2008), who analyzed the seasonal variations of the vertical profiles in the Bohai Sea.

The model results, however, show some discrepancies compared to the atlas. Although the model reproduces the summer saline stratification, it is weaker than in the atlas. Nonetheless, Li et al. (2015) reported that the summer salinity stratification in the Bohai Sea is possibly weaker than in the atlas, with an observed top-to-bottom salinity difference of 0.6 PSU. The modeled salinity stratification in summer is weaker compared to the atlas, which is possibly caused by the vertical mixing setting with the used

TKE closure scheme, and the high setting of vertical diffusivity in the model. In the north part of the transect, which corresponds to the northern Liaodong Bay, a negative salinity bias is visible compared to the atlas. In addition to the reasons mentioned in section 3.2.2, inaccuracies in the ETOPO1 bathymetry, especially in the low water depth region seen from Fig. 6 and Fig. 7, may also cause these underestimations.

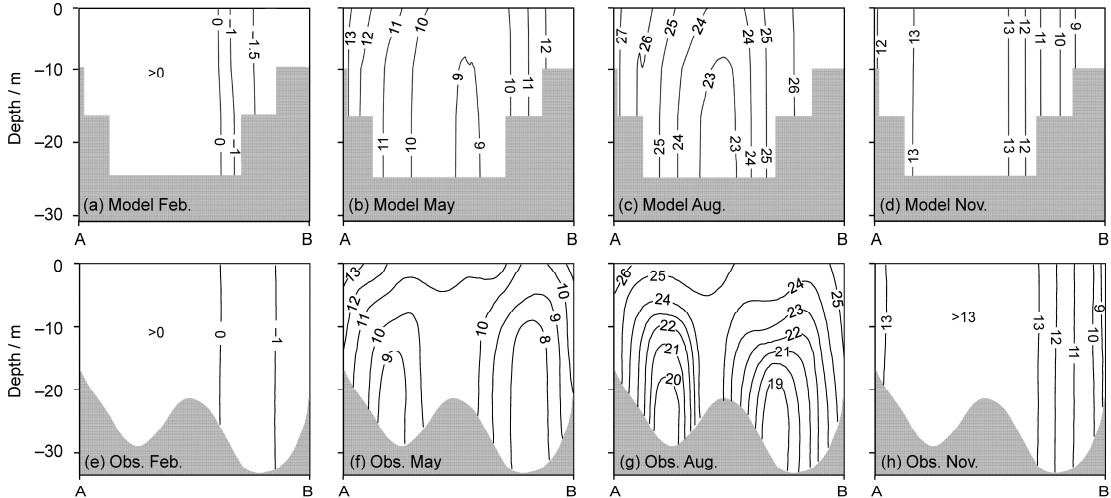

**Figure 6: Comparison of vertical profiles of water temperature (°C) along with transect AB (locations shown in figure 1) between NEMO-Bohai (a-d) and the atlas (Chen, 1992) (e-h) in February, May, August, and November.**

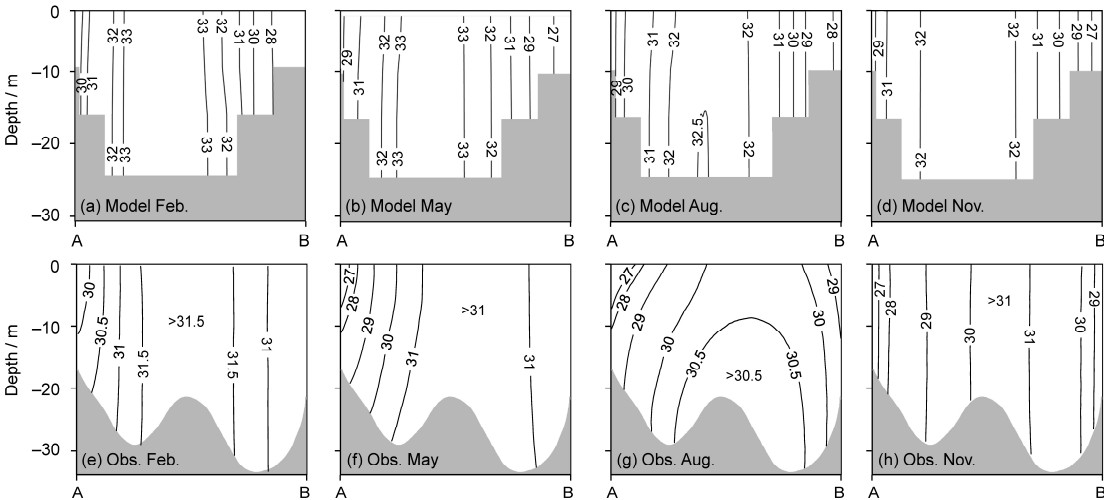

**Figure 7: Similar to figure 6 but for salinity (PSU).**

### 3.3 Sea ice

Sea ice is the focus of our study and is, in this section, directly validated against observations with respect to the area, thickness, volume, and their variations.

#### 3.3.1 Sea ice area

To evaluate the performance of NEMO-Bohai in simulating the characteristics of sea ice area, a series of variables were validated, including the daily sea ice area (DSIA), annual maximum sea ice area (AMSIA), and spatial patterns. Fig. 8 shows that the model-estimated DSIA agrees well with the satellite-derived observations ($r = 0.85$, $p < 0.01$). In addition to the good agreement of inter-annual variability,

the occurrence dates of the AMSIA are quite close to the observed dates from 1996 to 2017, as listed in

Table 4. The percentage of the occurrence dates of the AMSIA less than 3 days accounted for 72.7%, and

the mean absolute error is 3.5 days. In 2013, the maximum sea ice area was simulated on February 9,

2013, which is only one day later than reported (North China Sea Branch of State Oceanic Administration,

2013). However, the date retrieved from MODIS was determined on January 17 as the data on February

8, 2013 was missing due to severe cloud cover (Yan et al., 2017). The largest difference happened in the

winter of 2010 with 18 days earlier than the observations. Specifically, DSIA exhibited a bimodal

distribution with two remarkable peaks in the winter of 2010. Satellite showed that the AMSIA happened

on February 12, followed by the second maximum on January 26, while NEMO-Bohai simulated AMSIA

was on January 26, followed by the second maximum on February 12. It is worth noting that the satellite

shows the largest DSIA happened in the winter of 2010, while NEMO-Bohai simulated DSIA reached

the maximum in 2011. On the contrary, the modeled occurrence date of the AMSIA was six days later

than the observed date (February 7) in 2008.

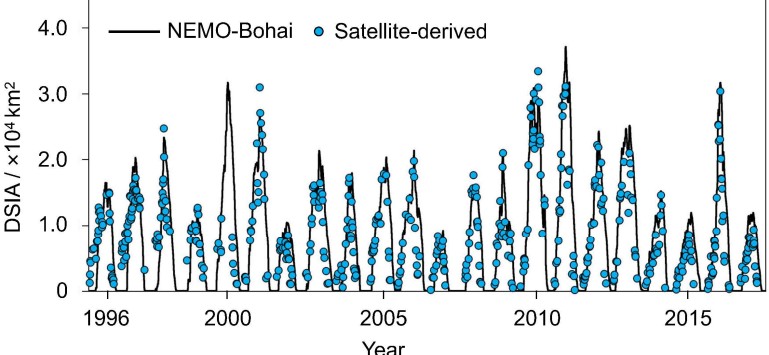

**Figure 8: Comparison of daily sea ice area (DSIA) between NEMO-Bohai simulations (black line) and satellite-derived data (blue circles) from 1996 to 2017.**

**Table 4: Comparison of the occurrence dates of the annual maximum sea ice area between the observations and model simulations.**

| Year | The occurrence dates of the annual maximum sea ice area | | | Year | The occurrence dates of the annual maximum sea ice area | | |
|------|----------|------------|-------|------|----------|------------|-------|
| | **Observed** | **NEMO-Bohai** | **Error** | | **Observed** | **NEMO-Bohai** | **Error** |
| 1996 | 2.10 | 2.10 | 0 | 2007 | 2.1 | 2.2 | +1 |
| 1997 | 2.3 | 2.3 | 0 | 2008 | 2.7 | 2.13 | +6 |
| 1998 | 1.28 | 1.25 | −3 | 2009 | 1.26 | 1.26 | 0 |
| 1999 | 2.3 | 2.4 | +1 | 2010 | 2.13 | 1.26 | −18 |
| 2000 | 2.8 | 2.3 | −5 | 2011 | 1.30 | 1.31 | +1 |
| 2001 | 2.7 | 2.7 | 0 | 2012 | 2.2 | 2.9 | −7 |
| 2002 | 1.25 | 1.24 | −1 | 2013 | 2.8 | 2.9 | +1 |

| 2003 | 2.2 | 1.30 | −3 | 2014 | 2.12 | 2.11 | −1 |
| 2004 | 2.5 | 2.7 | +2 | 2015 | 2.4 | 2.9 | +5 |
| 2005 | 2.19 | 2.21 | +2 | 2016 | 2.2 | 2.3 | −1 |
| 2006 | 2.8 | 2.9 | +1 | 2017 | 2.12 | 2.11 | −1 |

The comparison of the spatial distribution of sea ice from the NEMO-Bohai simulation and remote sensing inversion during the freeze, severe freeze, and thaw periods in light ice year (2007), normal ice year (2009), and heavy ice year (2010) is shown in Fig. 9. The simulated spatial distributions reflect general characteristics of sea ice evolution in the Bohai Sea except for the bias at the ice edge zones. Sea ice is mainly located in Liaodong Bay in light and normal ice years with extension to Bohai Bay and Laizhou Bay in heavy ice years. It is worth mentioning that NEMO-Bohai has well reproduced the development cycle of the sea ice but with a relatively slow thawing process.

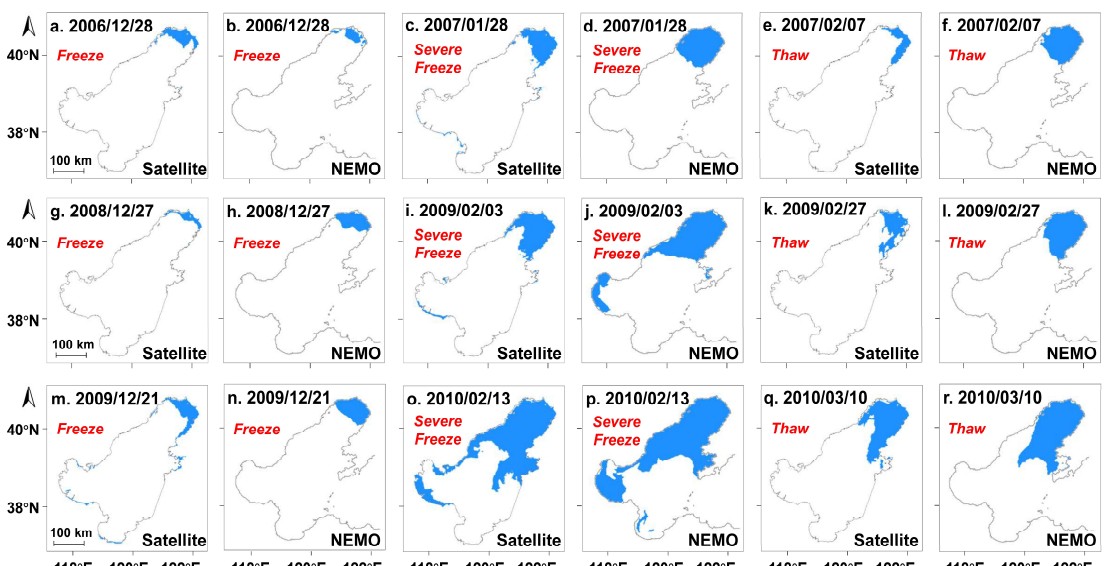

**Figure 9: Comparison of the spatial distribution of sea ice from NEMO-Bohai simulation and remote sensing inversion in freeze, severe freeze, and thaw periods for light (a-f), normal (g-l), and heavy (m-r) ice years.**

### 3.3.2 Sea ice thickness and volume

Fig. 10 shows that the modeled sea ice thickness based on NEMO-Bohai mostly lies in the range of in-situ observations with a slight overestimation. The mean relative bias of sea ice thickness between the simulations and observations is 4.6 cm ($n = 37$). Other studies also revealed that sea ice thickness is overestimated by NEMO in regional studies, such as in the Baltic Sea (Tedesco et al., 2017) and the Canadian Arctic Archipelago (Hu et al., 2018). When comparing ice thickness, the discrepancy of spatial scales makes it difficult to directly compare the modeled average of grids (~2 km) to the exact location

in-situ measurement as sea ice thickness varies significantly in the Bohai Sea, especially in the floating

ice zone.

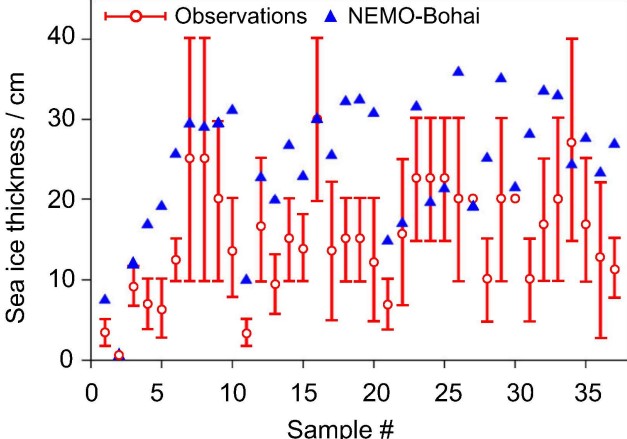

**Figure 10: The comparison of sea ice thickness between the simulated and observation data in the Bohai Sea. The red vertical bars represent the range of the observations, while the blue triangles denote the simulations based on NEMO-Bohai.**

Sea ice volume is defined as the total ice over the whole Bohai Sea, calculated through sea ice concentration multiplied by ice thickness in all grids. As demonstrated in Fig. 11, modeled daily sea ice volume is in reasonable agreement with satellite-derived data between 1996 and 2017. The modeled sea ice volume is higher than the observed, with the mean relative bias of $15.1 \times 10^8$ m$^3$ (RMSE=$20.3 \times 10^8$ m$^3$). The correlation coefficient between simulations and satellite-derived data is relatively high ($r = 0.80$,

$p < 0.01$, $n = 432$), indicating reasonable agreement at daily sea ice volume variations. In 2010, NEMO-Bohai simulations reproduced the range of $0.1{\sim}73.0 \times 10^8$ m$^3$, which agrees with the range of $4.4{\sim}63.0 \times 10^8$ m$^3$ reported by Liu et al. (2013) based on AVHRR satellite images. However, it is noticeable that there is a lag in modelling sea ice formation during the early freezing period compared to satellite observations.

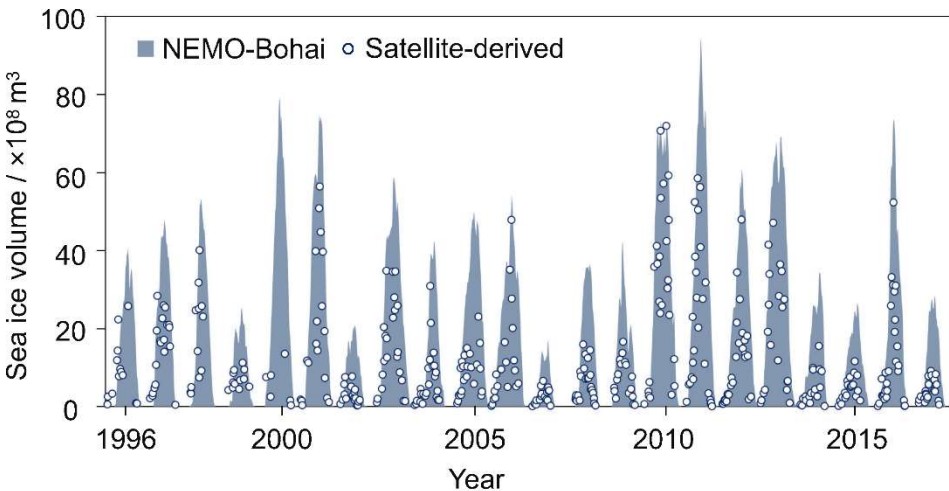


**Figure 11: Comparison of daily sea ice volume between satellite-derived data (circles) and NEMO-Bohai simulations (grey bars) from 1996 to 2017.**

Results of model comparison with in-situ data and satellite-derived data confirm the robustness of the developed model, which allows us to use it in a more detailed evaluation of the spatial and temporal

changes of Bohai Sea ice and study the continuous processes of freezing and melting at daily scales.

## 4. Initial results and applications

### 4.1 Ocean

#### 4.1.1 Sea surface temperature

Fig. 12 shows the monthly seasonal cycle of average SST in the Bohai Sea. There is an obvious

'warm tongue' starting from the Bohai Strait to the central Bohai Sea in winter (December, January, and February) due to the warm current from the Yellow Sea. It turns into a 'cold tongue' in summer due to faster temperature increases in shallow coastal waters as the influence of the warm current from the Yellow Sea also weakens. During January and February, the SST is below 0 °C in all three bays. There are large areas with SSTs below –1 °C in Liaodong Bay, the coldest sea in China during the winter (Ju

and Xiong, 2013). This extremely cold water provides an ideal platform for sea ice formation.

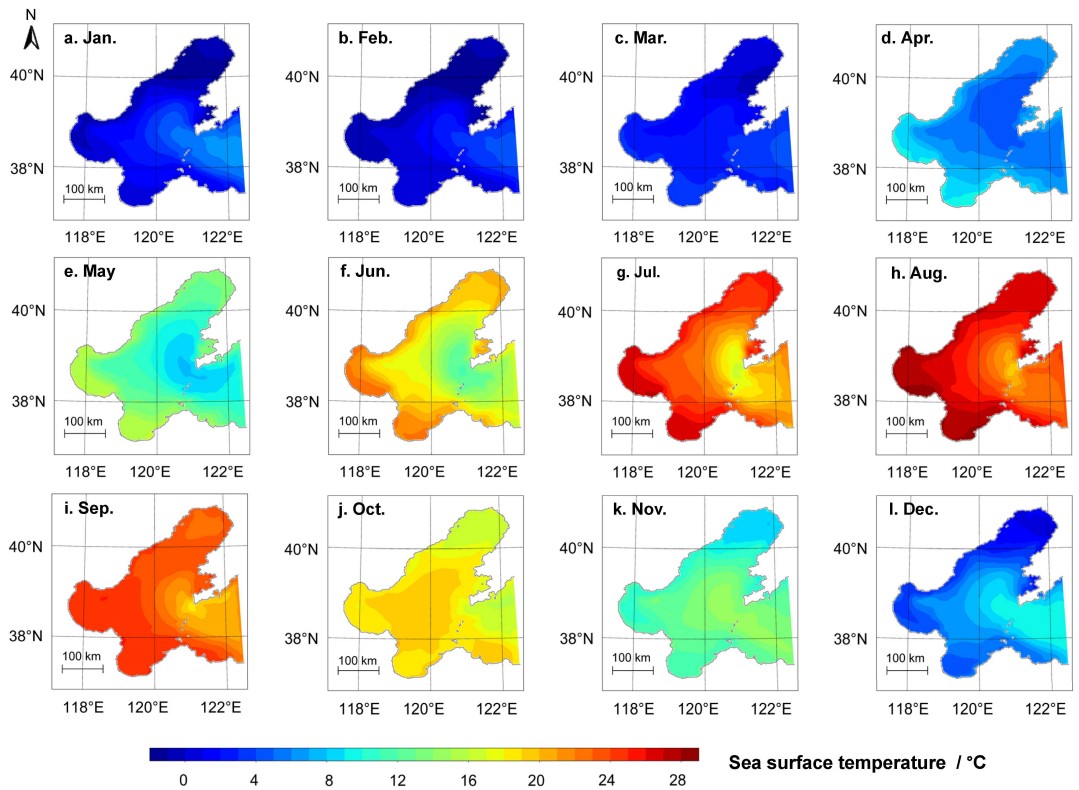

**Figure 12: Spatial distributions of multi-year monthly average sea surface temperature in the Bohai Sea for the period of 1996-2017.**

### 4.1.2 Sea surface salinity

As shown in Fig. 13, except for estuaries, the salinity isolines of the Bohai Sea are roughly parallel to the coast as the salinity in the coastal water is generally low due to river runoff. The average salinity of the whole Bohai Sea is low in summer (lowest in August at 29.5 PSU) and high in winter (highest in February at 30.0 PSU). The Bohai Strait in the east, which connects the Bohai Sea with the Yellow Sea, exhibits the highest salinity in the entire Bohai Sea, with an average of about 32.8 PSU.

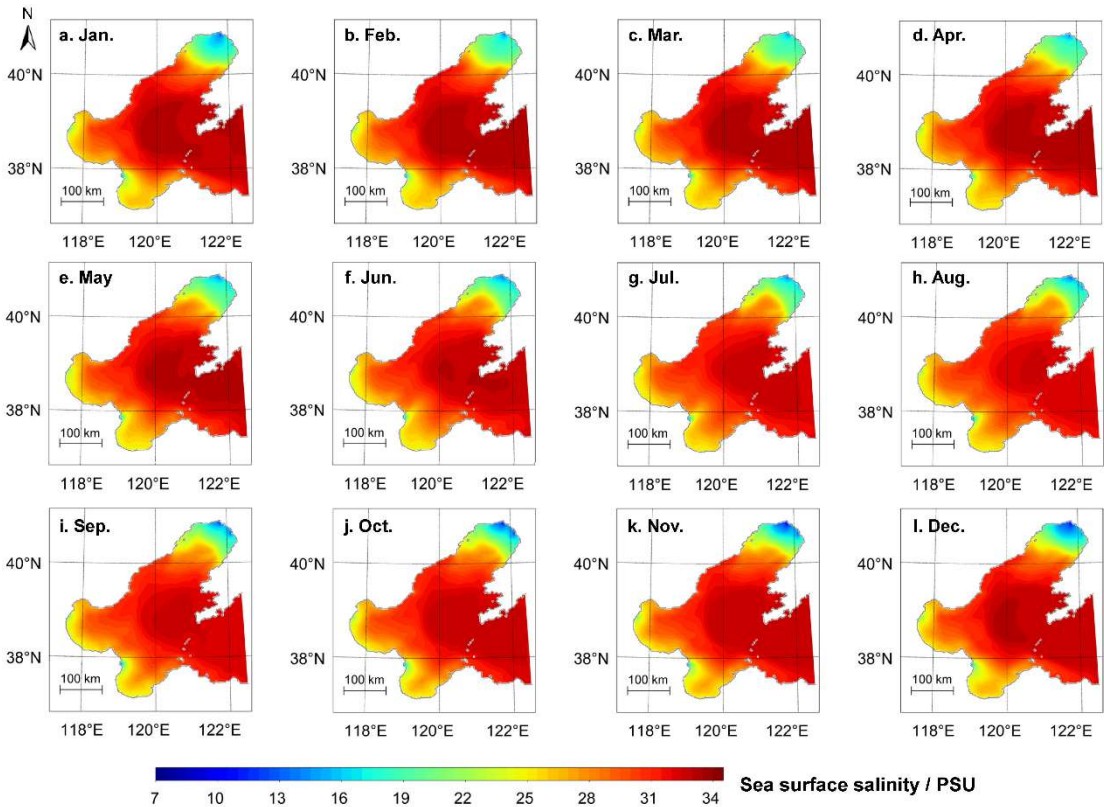


Figure 13: Spatial distributions of multi-year monthly average sea surface salinity (PSU) in the Bohai Sea during 1996-2017.

## 4.2 Sea ice

### 4.2.1 Temporal variation of Bohai sea ice

The variation of annual average and maximum sea ice area in the Bohai Sea based on NEMO-Bohai from 1996 to 2017 are shown in Fig. 14a and Fig. 14b, respectively. The annual average sea ice area (AASIA) is averaged by the DSIAs during the ice period for each year. The sea ice area exhibits apparent interannual and decadal variability in the study period. The mean AASIA during 1996-2017 is $1.07 \times 10^4$ km$^2$, accounting for 14.3% of the total sea area. 2010/11 and 2011/12 winters exhibited high AASIA

values of $1.71 \times 10^4$ km$^2$ and $1.73 \times 10^4$ km$^2$, covering 22.8% and 23.1%, respectively. The minimum AASIA reaches $0.54 \times 10^4$ km$^2$ (7.2% coverage) in 2007 winter, only accounting for half of the multi-year average AASIA. Analogously to AASIA, AMSIA is the maximum of the DSIAs during the period for each year. The mean AMSIA during 1996-2017 is $2.07 \times 10^4$ km$^2$ (27.6% coverage), and the AMSIA values in 2001/02, 2010/11, 2011/12 and 2016/17 winters are exceptionally high, larger than $3.00 \times 10^4$

km$^2$ (40% coverage). No significant trend in AASIA was observed, while AMSIA showed a non-significant increasing trend with $0.1 \times 10^4$ km$^2$ decade$^{-1}$ ($r = 0.08$, $p = 0.72$) from 1996 to 2017. In addition,

a strong positive correlation between AASIA and AMSIA ($r = 0.94$, $p < 0.01$) was found.

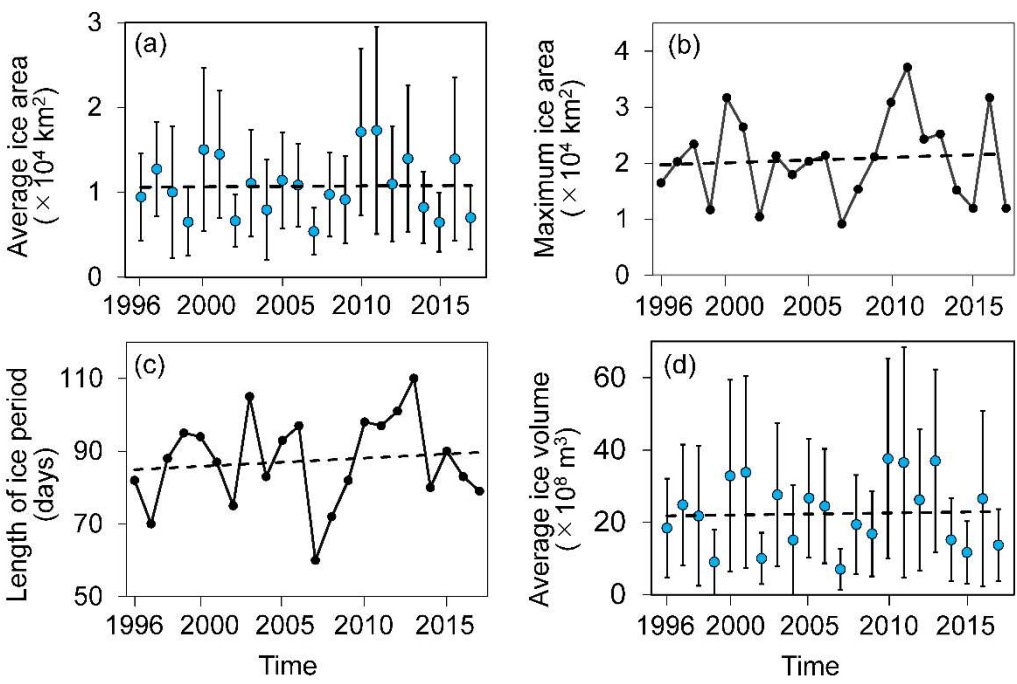

**Figure 14: The annual average sea ice area (a), the annual maximum sea ice area (b), the simulated length of ice period (c), and the annual average sea ice volume (d) in the Bohai Sea from 1996 to 2017. The blue dots present the annual average values, and the error bars stand for one standard deviation. The dotted lines represent 22 a-trends obtained through the linear fitting.**

The length of the ice period is defined as days with a sea ice area greater than 100 km$^2$ from December to March. It varies from 60 to 110 days during 1996-2017, as shown in Fig. 14c. The mean length of the ice period is 87 days, with a standard deviation of 12 days, and there is a slight increasing but statistically insignificant trend ($r = 0.12$, $p = 0.59$). The annual average sea ice volume (AASIV, defined as the average daily sea ice volume in the ice period for each year) from 1996 to 2017 is 2.24 billion m$^3$. As shown in Fig. 14d, no significant trends can be found for AASIV during the study period. The highest value of AASIV appeared in 2010, which was 3.76 billion m$^3$, and the lowest value appeared in 2007 with 0.7 billion m$^3$, accounting for only 18.6% of the ice volume of the highest year. The strong interannual variability of sea ice volume is determined by the strong interannual variability of sea ice area ($r = 0.96$, $p < 0.01$). Interestingly, it can be noticed that the ice period is not highly correlated with sea ice area or thickness, implying complicated processes of sea ice forming, growth, and thaw in the Bohai Sea.

The climatological seasonal cycles of the sea ice area and volume in the Bohai Sea averaged over 1996-2017 show unimodal variations (Fig. 15). Ice usually starts to form in mid-December and reaches

the maximum in early February. Then it starts to melt until the Bohai Sea becomes ice-free by mid-to-late March. The climatological mean of the length of the ice period is about three and a half months, and the freezing period (~8 weeks) is longer than the thawing period (~7 weeks). The asymmetrical process is related to changing rates of air temperature in freezing and thawing periods (Yan et al., 2020). Similarly, the seasonal variation of sea ice volume is an asymmetrical process. It is worth mentioning that the standard deviations of sea ice area and volume during the freezing period are pretty large, showing that the Bohai Sea ice has a large inter-annual variability.

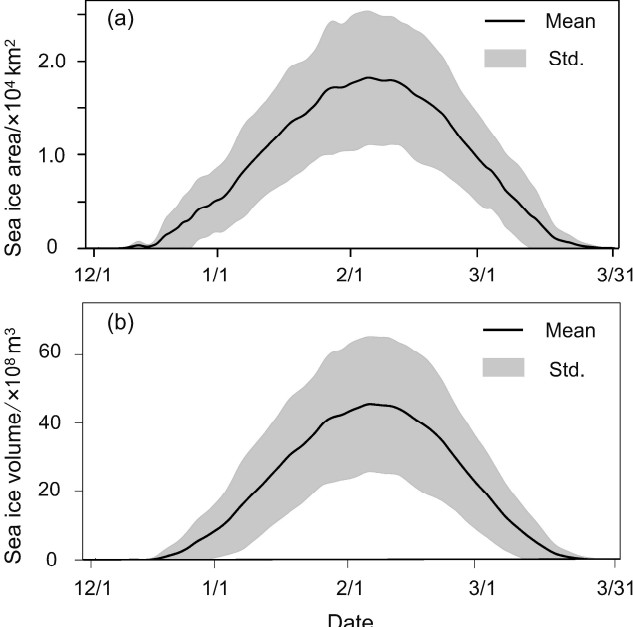

**Figure 15: Seasonal cycles of daily sea ice area (a) and sea ice volume (b) in the Bohai Sea. The lines show the daily averages, and the shading denotes one standard deviation. The seasonal cycles are calculated for the periods 1996–2017.**

**4.2.2 Spatial variation of Bohai sea ice**

This section will mainly discuss the spatial features of Bohai Sea ice. As shown in Fig. 16, there are substantial seasonal and spatial differences in sea ice concentrations. To be specific, seawater freezes first in Liaodong Bay, next in Bohai Bay, followed by Laizhou Bay, showing significant variation with latitude. On the contrary, the melting process happens exactly in reverse order. Similar to the aforementioned temporal changes in the last section, the sea ice concentration reaches its peak in February. Seawater in Liaodong Bay freezes most severely, and the high concentration zone moves from the north in January to the east in February, where it survives the longest until March.

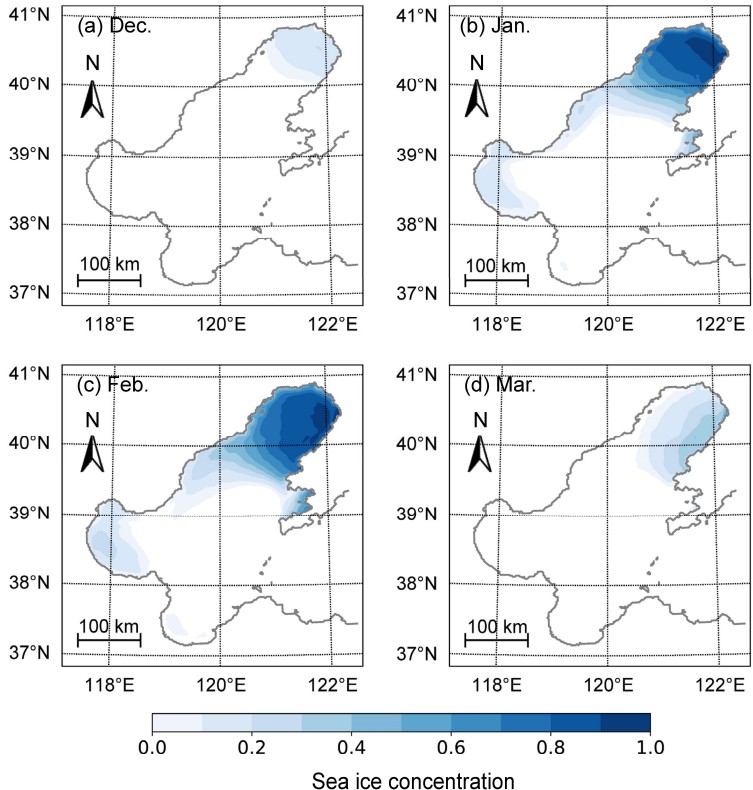

**Figure 16: Climatological monthly mean sea ice concentration in the Bohai Sea during 1996–2017, calculated as the average of all daily sea ice concentrations for each month during the ice period.**

As shown from Fig. 17, sea ice thickness shows quite similar seasonal and spatial features with sea ice concentration. The monthly mean Bohai Sea ice thickness simulated by NEMO-Bohai usually reaches its maximum in February, with a monthly mean thickness of 16.9 cm, following the second-highest monthly mean thickness of 15.8 cm in January. Sea ice is thicker on the east coast of Liaodong Bay than that on the west coast in January and February. The maximum sea ice thickness appears near the Bayuquan on the east coast of Liaodong Bay, where the thickness reaches up to ~60.0 cm in February.

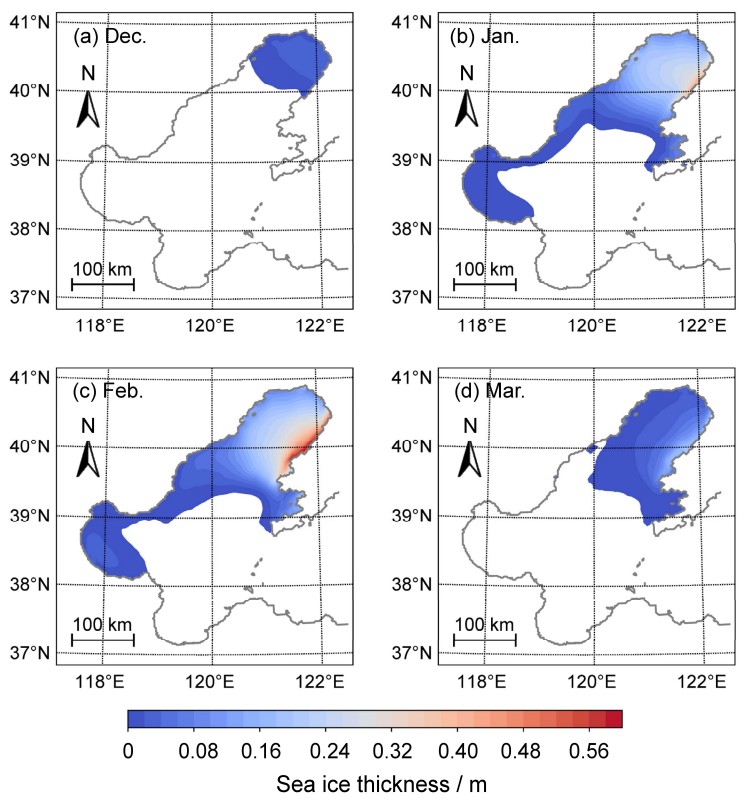

**Figure 17: Similar to figure 16 but for sea ice thickness.**

According to the guideline for risk assessment and zoning of sea ice disaster issued by the State Oceanic Administration in 2016, a high-risk level is reached when sea ice thickness becomes greater than 25 cm, while the risk level is low when sea ice thickness is lower than 10 cm. When sea ice thickness is

515 between 10 cm and 25 cm, the risk level is moderate. Accordingly, in this study, the thresholds for moderate-risk and high-risk levels of sea ice disaster are set at 10 cm and 25 cm, respectively. In Fig. 18, sea ice risk maps are calculated based on daily sea ice thickness from 1996 to 2017, and they clearly show that the high-risk area is mainly located at Liaodong Bay, with the highest risk area in the eastern Liaodong Bay. Thus, the seas around Yingkou, Bayuquan, and Wafangdian bear high exposure to sea ice

disasters.

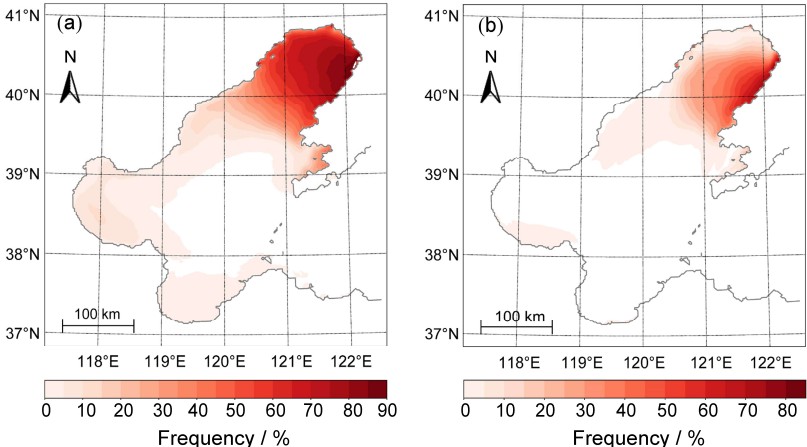

**Figure 18: Frequency distribution of daily sea ice thickness larger than 10 cm (a) and 25 cm (b) in the Bohai Sea during 1996–2017.**

As the drift ice is a major component in Bohai Sea ice, studying its motion characteristics is also essential. As shown in Fig. 19, the direction is mainly northeast-southwesterly, and the drift exhibits a high spatial variability. The zone with the highest sea ice velocity (~0.15 m s$^{-1}$ and primarily directed to the southwest) is visible at the southeast edge of Liaodong Bay, next to the high concentration and thickness area. In February, the velocity in the northern Liaodong Bay is significantly lower than that in the southern part near the ice edge.

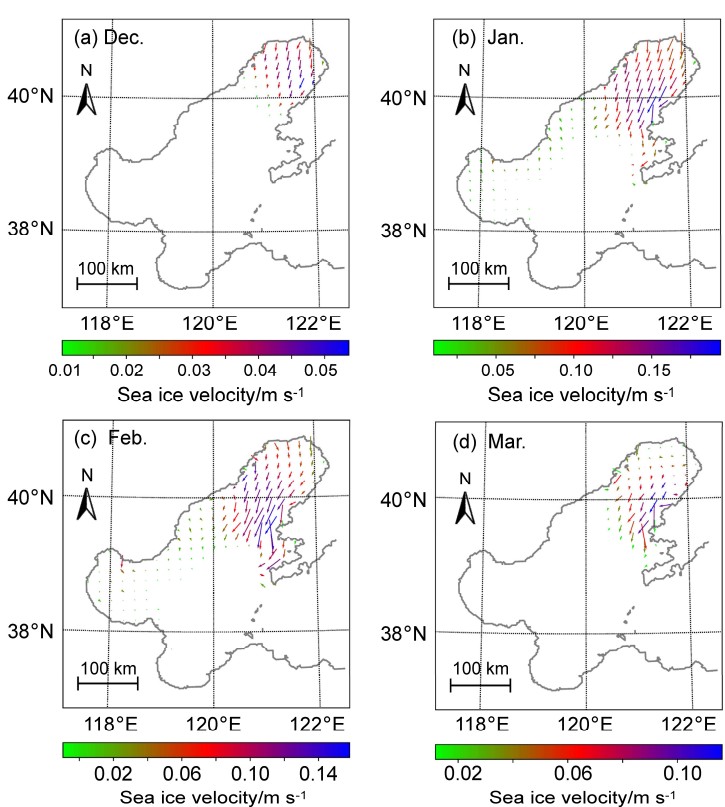

**Figure 19: Similar to figure 16 but for sea ice velocity.**

**4.2.3 Correlations between sea ice and climate factors**

The synoptic forcing may play an essential role in the changes of Bohai Sea ice which is primarily influenced by the Eurasian continental climate. To explore the potential regional climate drivers on the evolution of sea ice, correlations between sea ice and various indices based on detrended air temperature, air pressure, wind speed, and precipitation were examined. These indices were calculated based on daily meteorological measurements during the winter (DJFM) from 1996 to 2017 at 12 weather stations (Dalian, Wafangdian, Xiongyue, Yingkou, Jinzhou, Xingcheng, Suizhong, Qinhuangdao, Laoting, Tanggu, Huanghua, Dongying) surrounding the Bohai Sea. These measurements were obtained from the China Meteorological Data Service Center (http://data.cma.cn/en). As shown in Fig. 20a, a strong negative correlation ($r = -0.78$, $p < 0.01$) between the detrended time series of AASIA and the mean winter air temperature (MWAT) is observed. Similarly, the detrended AASIV exhibits even a stronger negative correlation ($r = -0.87$, $p < 0.01$, Fig. 20e) with MWAT, suggesting that MWAT is an essential regional climate factor for sea ice changes. The detrended Bohai Sea ice area exhibits a weak but insignificant positive correlation ($r = 0.28$,$p = 0.21$, Fig. 20b) with the mean winter relative humidity (MWRH). The correlation between the detrended AASIV and the detrended MWRH is also found to be weak and insignificant ($r = 0.36$,$p = 0.10$, Fig. 20f). In addition, the correlations between the detrended AASIA and the detrended time series of mean winter wind speed (MWWS, $r = 0.13$,$p = 0.56$, Fig. 20c) and mean winter air pressure (MWAP, $r = 0.04$,$p = 0.85$, Fig. 20d) were found to be very weak and insignificant. Similar very weak and insignificant correlations were also found between AASIV and MWWS/MWAP (see Fig. 20g, 20h). The above correlation analysis indicates that the local temperature is the controlling factor for the sea ice evolution in the Bohai Sea.

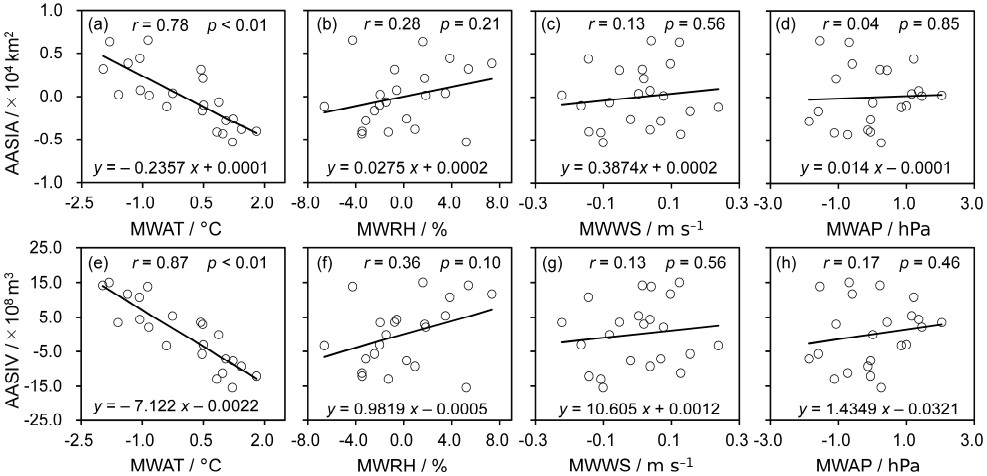

**Figure 20: Pearson correlations between annual average sea ice area (AASIA) (a-d), annual average sea ice volume (AASIV) (e-h) and mean winter air temperature (MWAT), mean winter relative humidity (MWRH), mean winter wind speed (MWWS), mean winter air pressure (MWAP) through detrended analysis for the period 1996–2017. The climate parameters are calculated based on the measured daily meteorological data at 12 meteorological stations surrounding the Bohai Sea during the winter (DJFM).**

Fig. 21 illustrates the spatial correlations between daily sea ice concentration with vertically

integrated ocean heat/salt content from the surface to the bottom of the mixed layer during the ice period

for each grid point. Ocean heat content strongly correlates with sea ice concentration in Liaodong Bay

and moderately correlates near the coastal areas. The spatial pattern corresponds quite nicely with

climatological monthly mean sea ice concentration (shown in Fig. 16), with the highest values appearing

at the east of Liaodong Bay. On the other hand, a very weak positive or negative correlation between

daily sea ice concentration and integrated ocean salt content was found in most sea areas, indicating a

more complicated relationship throughout the whole Bohai Sea. Therefore, the interannual variability of

sea ice is more dominated by ocean heat content than salt.

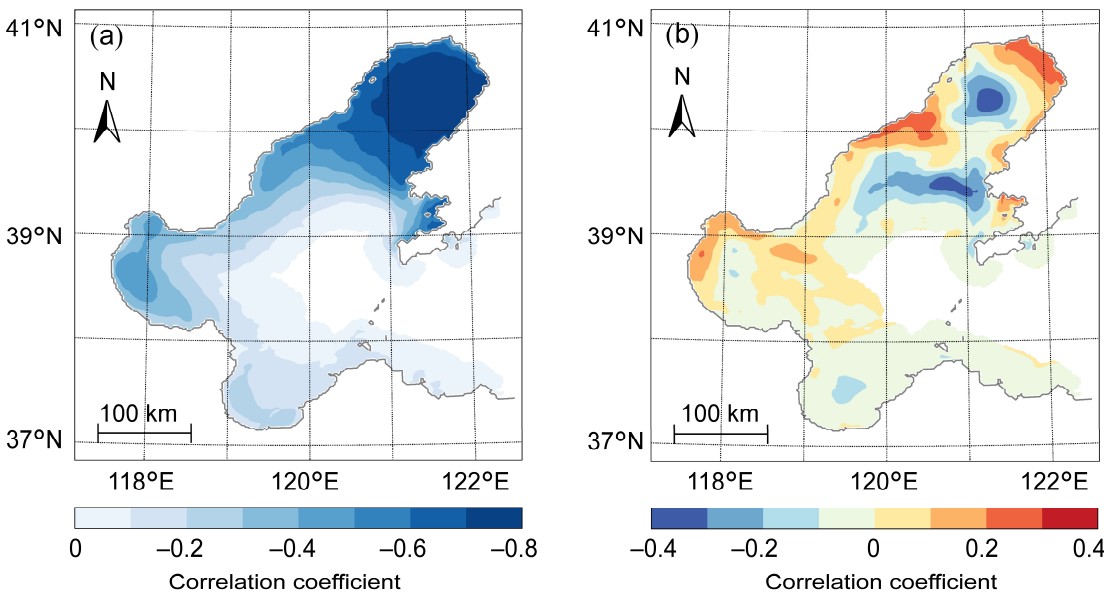

**Figure 21: Spatial correlation between daily sea ice concentration in the Bohai Sea and (a) vertically**
**integrated ocean heat content, and (b) vertically integrated ocean salt content from the surface to the bottom of the mixed layer during the ice period (DJFM).**

**5. Conclusion and perspectives for the modelling platform**

In this study, we provided a detailed description of NEMO-Bohai, a newly developed setup of an ocean-ice model for the Bohai Sea. The primary intent of our study was to test how NEMO-Bohai represents the ocean characteristics and sea ice properties.

Comparisons with observational data confirm that NEMO-Bohai is able to reproduce reasonably well the ocean properties, including ocean surface information (e.g., SSH, SST, SSS), currents, and temperature and salinity stratification. However, the ongoing development of the NEMO ocean engine, soon providing an updated version with the inclusion of wetting and drying processes, could improve the model performance in terms of sea surface height in shallow areas (Hordoir et al., 2019), especially for the Bohai Sea with an average depth of only 18 m. A higher-precision bathymetry could also improve the SSH performance. In this study, the monthly climatology of coastal runoff flux from Dai et al. (2009) was recorded only for 1948–2004, different from the study period. More importantly, the river runoff is based on climatological estimates without interannual variability, and we assume that the river salinity is set to 0 PSU and the runoff temperature is the same as the ocean surface. Major future development would be using gauge records with observations of temperature and salinity assimilated into the NEMO-Bohai model, which would reduce uncertainties in temperature and salinity. Furthermore, a possible implementation of multiple embedded methods (Hvatov et al., 2019; Schwarzkopf et al., 2019), such as Global Ocean-West Pacific Ocean-East China Sea-Bohai Sea rather than current Global Ocean-Bohai Sea direct nesting, should be investigated in the future. In addition, in order to carry out more accurate estimation of vertical mixing, it is worth implementing the experiments of turbulent vertical mixing options (Reffray et al., 2015) for the Bohai Sea for further development of NEMO-Bohai. Moreover, interactive feedback from the Bohai Sea to the global ocean could also be considered in a two-way coupling method compared to current one-way nesting.

For the sea ice component, NEMO-Bohai reproduced satisfactorily the seasonal and interannual variabilities of sea ice area compared to the satellite remote sensing data for the period 1996-2017. The modeled dates of the annual maximum sea ice area were 0.9 days earlier than the observed ones. Spatially, the simulation results realistically reflect the main characteristics of Bohai Sea ice evolution compared to the satellite data. Therefore, NEMO-Bohai can reliably be used to provide ice information during the dates without satellite-derived data. Nevertheless, it is worth mentioning that the simulated sea ice area

tends to be somewhat overestimated, which is also reported for other seas in earlier NEMO-related publications (Blockley et al., 2014; Massonnet et al., 2011; Rjazin et al., 2019). In spring, the NEMO-Bohai melting process is delayed, particularly in the land-fast ice zone in the Eastern Liaodong Bay. It is likely that thick ice melts slower away than thinner observed ice. The applied parameterization of the surface albedo may cause a slow melting process, which does not take realistically into account the relevant physical processes, such as surface melt ponds, as the surface albedo continues to decrease until sea ice is completely disappeared (Mortin et al., 2016). Thus, regional atmospheric forcing data with higher accuracy and resolution can be used for further development. In addition, the space discrepancy between modeled sea ice thickness and extremely limited in-situ observations makes their comparison difficult and introduces significant uncertainties.

In conclusion, NEMO-Bohai can simulate ocean and sea ice properties with reasonable skill in a broad spatiotemporal context, especially in terms of seasonal evolution and long-term interannual variations of sea ice. This finding implies that NEMO-Bohai well complements the discontinuous satellite data in sea ice hazard risk analysis. Therefore, NEMO-Bohai is a valuable tool for long-term ocean and ice simulations and climate change studies.

**Code and data availability**

The NEMO-Bohai is built upon the standard NEMO code (NEMO 4.0 beta, revision 10226). The reference code is available at the NEMO website (http://www.nemo-ocean.eu/) through the following link: http://forge.ipsl.jussieu.fr/nemo/svn/NEMO/trunk (last access: May 20, 2021). The parameters and configuration files running the NEMO-Bohai, such as namelists, bathymetry, boundary coordinates, and definition files, forcing data etc., together with the scripts to run the model, are available at https://doi.org/10.5281/zenodo.4892454. The forcing data for the year 2010 are included in the Zenodo archive.

**Author contributions**

YY, WG, and PU designed the study. YY and AG ran the experiments. YY and YX analyzed the model and observational data. YY and PU wrote the manuscript. All authors provided scientific input.

**Competing interests**

The authors declare that they have no conflict of interest.

**Acknowledgements**

This study was financially supported by the National Natural Science Foundation of China (Nos. 41977406, 41571510), the Academy of Finland (grant 322432), the China Scholarship Council (No. 201806040130), 111 project (No. B20011), and the National Key Research and Development Program of China (No. 2017YFA0604903). This work was granted access to the HPC resources of CSC - IT

Center for Science. We gratefully thank Juha Lento, Robinson Hordoir, Yongmei Gong, Yu Zhang, Xiaoqiao Wang for sharing the scripts for input data processing.

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
