# Peer review of "NEMO-Bohai 1.0: a high-resolution ocean and sea ice modelling system for the Bohai Sea, China"

_Geoscientific Model Development, 2021_

## Author Comment (AC2)

**Response to Reviewer 1**

We are grateful for your insightful comments on our manuscript and have revised the manuscript accordingly. All your comments (in blue) are addressed (in black) point by point, with new text added in the manuscript highlighted in *Italic*.

This article presents a new NEMO based configuration for the Bohai Sea, which is a small semi-enclosed and very shallow area in North Eastern China, located at the latitude of Beijing. It is to my knowledge the smallest area covered by a NEMO configuration. The authors particularly focus on the sea-ice characteristics of the area.

The concept is interesting, Beijing is located at a low latitude (Valencia in Spain, in European standards) where the presence of sea-ice is possible only due to a combination of cold continental winters, very shallow waters and low salinity in the Bohai Sea. This makes the Bohai Sea very sensitive to climate change from the atmosphere but also from the ocean point of view.

From a general perspective, I think the article in its present state does not explore such aspects thoroughly enough, which is a bit sad because I believe this aspect to be really interesting. Therefore I think this article could be published in GMD after major revision.

We greatly appreciate the reviewer for the interest in our work. We have revised the manuscript in this regard to tell a more comprehensive story about Bohai Sea ice. In particular, the ocean part has now been added into the discussion covering aspects related to the temperature and salinity stratification, currents, and water volume exchanges with the Yellow Sea, and etc., in addition to sea ice itself.

Also there are editing problems in pages 6-10 of the article, the authors made an online comment on this but only for an array, which does not improve the readability at all. I strongly advise to correct this to make the work of reviewers nicer to do.

Thank you for your comment. We recognize that the garbled items are due to technical issues that affect the readability and have been corrected in this revised version.

Most important comments:
- First, could you present the Bohai Sea a bit better. Especially we would like to understand what makes there is sea ice in the Bohai Sea. Basically there would not be any sea ice in the deep ocean at such a latitude. Is there sea ice in the Bohai Sea because it is very shallow or because of the low salinity or both? Basically, please provide a good hydrography of the Bohai Sea.

Yes, understanding the formation of sea ice in the Bohai Sea is a subject of great concern. Following your suggestions, we have detailed this part in the introduction:

*The formation of sea ice in the Bohai Sea mainly depends on the geographical environment and hydrometeorological characteristics (Ding, 1999). Specifically, the Bohai Sea is located in the continental shelf area, and the average water depth is only 18 m (Su and Wang, 2012), which indicates low oceanic heat content in winter. Following a northern continental climate, the Bohai Sea is affected by the cold Siberian air every winter, which causes the sea surface temperature of the Bohai Sea to be significantly lower than that at the same latitude (Zhang et al., 2016; Donlon et*

*al., 2012). In addition, the sea surface salinity of the Bohai Sea is about 30 PSU, which is the lowest in the entire coastal waters in China (Yan et al., 2020). It means that the Bohai seawater freezes before reaching the maximum density. Therefore, it even more easily convects and loses heat just before freezing.*

- The paper mostly deals with sea ice, and neglects the Bohai Sea circulation. Is it mostly barotropic and driven by tides or is there a contribution of baroclinic effects, is there haline or thermal stratification? Basically, if the low salinity of the Bohai Sea is essential to explain the sea ice cover, then it is crucial to know what are the processes that drive the salinity balance of the Bohai Sea. Additionally could you provide estimates of how much flow comes from the Yellow Sea and leaves, is the circulation mostly cyclonic or is it not even geostrophic perhaps? This point is important because we need to understand if the long term sea ice cover trends are only driven by atmospheric forcing, or if changes in ocean circulation and/or river runoff can also affect the sea ice cover through the long term changes of SSS. And of course, we want to know how the model compares with estimates of this circulation.

Thank you for your comment. We have detailed this part to a larger extent. A new section 3.2.3., including plotting the simulated currents at surface and lower layer and estimating the monthly mean water volume exchange at the Bohai Strait as suggested, was added as suggested. Further analysis of the vertical profile of temperature and salinity in later parts will help understand the thermal or haline stratification in the Bohai Sea.

**3.2.3 Sea current**

*The circulation in the Bohai Sea is mainly barotropic and results from the combined effects of tides and winds (Li et al., 2015). The simulated monthly mean current velocities at the surface and 16 m depth in February and August are shown in Fig. 5. The monthly mean current velocities are calculated based on hourly model output during August 2012 and February 2013. The figure shows that both the sea surface and 16 m depth current are usually less than 0.4 m s$^{-1}$. Due to the blocking effects of the bays, currents are weak at the head of the three bays, which is consistent with the observations by Chen et al. (1992). The surface currents in the Bohai Sea also show apparent seasonal variations. In August, relatively strong currents flow from the southeastern part of Liaodong Bay to the southern Bohai Strait with a solid outflow to the Yellow Sea. Specifically, the current field distribution at Bohai Strait shows that the inflow is mainly in the narrow channel in its northern part, which agrees with the observations of Wan et al. (2015). Also, Lin et al. (2011) suggested that persistent winds drive a cyclonic coastal current into the northern Yellow Sea, and one branch of the current enters the Bohai Sea at the northern Bohai Strait, which transports warm and saline water from the Yellow Sea.*

*The monthly mean water volume exchange at the Bohai Strait (see Fig.1) based on hourly model simulations during August 2012 and February 2013 are also calculated to evaluate the model's performance. The Bohai Sea water exchange with the Yellow Sea is weak due to its half-closed shape and a relatively independent circulation system. The model results show that the inflow from the Yellow Sea to the Bohai Sea in August reaches $6.9 \times 10^4$ m$^3$ s$^{-1}$, almost double than that in February ($3.5 \times 10^4$ m$^3$ s$^{-1}$), which lies in the range of $5 \times 10^3$ to $8 \times 10^4$ m$^3$ s$^{-1}$ indicated by Bian et al. (2016). Our results show that the outflow exists in both months (August: $8.3 \times 10^4$ m$^3$ s$^{-1}$; February: $6.5 \times 10^4$ m$^3$ s$^{-1}$), with a larger amount than the inflow. The net flow appears to be outflow both in winter*

*and summer, which is consistent with other model simulation results (Lin et al., 2002; Ji et al., 2019).*

[Figure]

***Figure 5: Simulated monthly mean current velocities at surface and 16 m depth in August 2012 and February 2013. The monthly mean current velocities are calculated based on the outputs with hourly intervals.***

- Although the paper lacks a proper presentation of the Bohai Sea hydrography, there is an extensive presentation of ORCA025 used at the OBCs. Could you just replace it with a simple reference?

This part has now been shortened as suggested. The extensive presentation of ORCA025 is replaced by the reference of Bernard et al. (2006). We have only kept the updated forcing datasets, which are often overlooked by model users and worth mentioning in this part.

- As mentioned before, the Nemo-Bohai settings are very difficult to read. But I understood you use the Blanke & Delecluse TKE turbulence scheme. This choice is a bit odd in such a configuration but perhaps does not matter too much if the tidal mixing kills any form of stratification, and therefore I would understand that this is not much of a concern. The background vertical diffusivity is also very high, but perhaps it does not matter too much for the same reasons. However, could you show some T/S profiles and comparison with data in some of the deepest areas? Having the right density profile close to the OBCs is an important feature to get the right amount of estuarine circulation. This part would be of course a lot easier to understand if you have first presented a proper hydrography of the Bohai Sea as mentioned before.

We have revised the garbled symbols to improve the readability of the revised manuscript. Indeed, we agree that comparing the temperature and salinity vertical profiles between the model and observations is very valuable. We have followed you and added the T/S profile figures (Fig. 6 and Fig. 7) accordingly, and relevant texts have been added to the new section 3.2.4:

NEMO-Bohai and observed water temperature and salinity profiles along the transect AB (see Fig.1) are shown in Fig. 6 and Fig. 7, respectively. Observations are from the atlas by Chen (1992), which is based on data from the 1950s to 1990s. The temporally closest 5-year period from 1995 to 2000 of NEMO-Bohai simulations was selected for model-observation comparisons. Common features are found both in the model and observations. The Bohai Sea waters are vertically well-mixed in autumn and winter, and they have a remarkable homogeneous vertical distribution for both temperature and salinity. In spring and summer, thermal stratification occurs with a significant cold-water core at depth, eventually eroded in autumn. As apparent in Fig. 6 and Fig. 7, the stratification in shallow coastal waters is generally homogeneous. Similar features were reported by Wang et al. (2008), who analyzed the seasonal variations of the vertical profiles in the Bohai Sea.

The model results, however, show some discrepancies compared to the atlas. Although the model reproduces the summer saline stratification, it is weaker than in the atlas. Nonetheless, Li et al. (2015) reported that the summer salinity stratification in the Bohai Sea is possibly weaker than in the atlas, with an observed top-to-bottom salinity difference of 0.6 PSU. The modeled salinity distribution along with transect AB during summer is possibly affected by the high vertical diffusivity. In the north part of the transect, which corresponds to the northern Liaodong Bay, a negative salinity bias is visible compared to the atlas. In addition to the reasons mentioned in section 3.2.2, inaccuracies in the ETOPO1 bathymetry, especially in the low water depth region seen from Fig. 6 and Fig. 7, may also cause these underestimations.

[Figure]

*Figure 6: Comparison of vertical profiles of water temperature (°C) along with transect AB (locations shown in figure 1) between NEMO-Bohai (a-d) and the atlas (Chen, 1992) (e-h) in February, May, August, and November.*

[Figure]

*Figure 7: Similar to figure 6 but for salinity (PSU).*

The lack of stratification in summer is possibly caused by the high setting of vertical diffusivity in the model, as you indicated, and sensitivity experiments with different turbulent energy closure schemes will be further studied in the future.

- The validation part of the ocean follows the weakness of the paper I think, it lacks some interest of the ocean. Please provide statistics when they are useful, especially for SSH: standard deviation of both model and observations, correlation, and mean square root error. And in this part you can also provide a comparison with S/T profiles which will allow to check the turbulence and the circulation. Having statistics on salinity or temperature is not really essential if we don't understand why, and there are only measurement stations along the coast.

Basically please extend the validation of the ocean with explanation of the biases rather than a purely descriptive approach.

Indeed, thanks to this remark. We have included more information on the validation part. Specifically, in terms of SSH, we have added extra validation results and a table (see Table 2) in section 3.2.1 with more information such as standard deviation, correlation, mean absolute errors, and root-mean-square error. As for the comparisons of S/T profiles, we have added the figure and text in the new section 3.2.4 accordingly.

To better explain the results of the model-observation comparison, we have expanded the explanation of the biases as suggested. For example, we have supplemented the possible reasons for SSH deviation in section 3.2.1 and addressed the reasons for the vertical profile discrepancy in section 3.2.4.

In section 4 come some results about hydrography, which we have to assume are correct based on a validation that is rather light.

We have added the comparisons of temperature and salinity vertical profiles between NEMO-Bohai and the atlas, as well as the validation of currents. In addition, more explanations for model biases are proposed in section 3. Our main finding is that the NEMO-Bohai reproduces the main variables of ocean and sea ice well.

The most interesting part of the paper comes in this section, but is not exploited at all. It is about the sea ice cover. Obviously there is no significant trend in sea ice cover or volume (or even a slight increase?), but could you relate this with trends in atmospheric forcing? In such a shallow region, the sea ice cover should be highly correlated with the mean winter air temperature. Is it the case or are there other factors? Could you plot trends in temperature and salinity in the Bohai Sea? You could integrate the heat and salt content to do that for example, in the mixed layer (unless it is always mixed to the bottom). The climatological cycles are interesting, but the interest of having a long integration is to see trends and not only in sea-ice cover, and to understand what drives the inter-annual variability.

We agree that sea ice is significantly related to atmospheric and possible oceanic forcing in such shallow water. It would be fascinating to understand what drives the lack of significant trends of sea ice cover or volume, so here we have added a new section 4.2.3 to explore the potential regional climate driving factors on the evolution of sea ice as suggested. As you indicated, the local temperature is the controlling factor for the sea ice evolution in the Bohai Sea. In addition, we have further defined the oceanic effects on the sea ice by investigating the spatial correlation between daily sea ice concentration and vertically integrated ocean heat/salt content (see newly added Fig. 21). Our main finding is that the interannual variability of Bohai Sea ice is more dominated by heat content than salt.

**Some other comments:**
- A general comment is that there are many English language mistakes, I had tried to pick them one by one and finally renounced. Please have the article checked from this point of view when submitting the revised version.

We have employed the English Language Editing service to improve the language presentation.

- Please check all the weird symbols in page 6-10.

Corrected. The certain terms in Nimbus Mono L Font are garbled, and we have revised these weird symbols accordingly in the revised manuscript.

- Please check all your units which are inconsistent, the right format for unit notation is for example $1.2 \text{ m}^3 \text{ s}^{-1}$
Meaning that there should not be any . $\text{m}^3$ and $\text{s}^{-1}$
nor any multiply or / sign. Salinity should be in PSU. Please check with Copernicus if they ask for the exponent to be 10 or just the "e" letter.

Thanks for your careful work on our manuscript. We have checked the mathematical notation and unit notation one by one and corrected them according to manuscript guidelines required by Copernicus Publications. Regarding the unit of salinity, we have replaced it with the unit of PSU as suggested.

- Line 349: "It is noticeable that modeled sea ice volume change is slightly later than the observed by satellite during early freezing period", please clarify

We have revised this line to clarify this point.
*It is noticeable that there is a lag in modelling sea ice formation during the early freezing period compared to satellite observations.*

The text has been revised.

**Reference**

Bernard, B., Madec, G., Penduff, T., Molines, J. M., Treguier, A. M., Le Sommer, J., Beckmann, A., Biastoch, A., Böning, C., Dengg, J., Derval, C., Durand, E., Gulev, S., Remy, E., Talandier, C., Theetten, S., Maltrud, M., McClean, J., and De Cuevas, B.: Impact of partial steps and momentum advection schemes in a global ocean circulation model at eddy-permitting resolution, Ocean Dynam, 56(5), 543-567, 2006.

Bian, C., Jiang, W., Pohlmann, T., and Sündermann, J.: Hydrography-physical description of the Bohai Sea, J Coast Res, 74, 1-12, 2016.

Chen, D. X.: Marine atlas of Bohai Sea, Yellow Sea and East China Sea: Hydrology, Ocean Press, Beijing, China, 1992.

Ding, D.W.: Introduction to sea ice engineering (in Chinese), Ocean Press, Beijing, 1999.

Donlon, C. J., Martin, M., Stark, J., Roberts-Jones, J., Fiedler, E., and Wimmer, W.: The operational sea surface temperature and sea ice analysis (OSTIA) system, Remote Sens Environ, 116: 140-158, 2012.

Ji, C. Z., Li, K., Yu, B., Dong, L., and Liu, Q. R.: The multi-time scale variations of water exchange across the Bohai Strait, Oceanol Limnol Sin (in Chinese), 50, 24-30, 2019.

Li, Y., Wolanski, E., and Zhang, H.: What processes control the net currents through shallow straits? A review with application to the Bohai Strait, China, Estuar Coast Shelf S, 158, 1-11, 2015.

Lin, X., Wu, D., Bao, X., and Jiang, W.: Study on seasonal temperature and flux variation of the Bohai Strait, Journal of Ocean University of Qingdao (in Chinese), 32(3), 355-360, 2002.

Lin, X., Yang, J., Guo, J., Zhang, Z., Yin, Y., Song, X., and Zhang, X.: An asymmetric upwind flow, Yellow Sea warm current: 1. New observations in the western Yellow Sea, J Geophys Res-Oceans, 116(C4), C04026, 2011.

Su, H. and Wang, Y.: Using MODIS data to estimate sea ice thickness in the Bohai Sea (China) in the 2009-2010 winter, J Geophys Res-Oceans, 117, C10018, 2012.

Wan, K., Bao, X., Wang, Y., Wan, X., Li, H., and Liu, K.: Barotropic current fluctuations coupled with sea level drawdown in Yellow and Bohai Seas, Chin J Oceanol Limn, 33(1), 272-281, 2015.

Wang, Q., Guo, X., and Takeoka, H.: Seasonal variations of the Yellow River plume in the Bohai Sea: A model study, J Geophys Res, 113, C08046, 2008.

Yan, Y., Uotila, P., Huang, K., and Gu, W.: Variability of sea ice area in the Bohai Sea from 1958 to 2015, Sci Total Environ, 709, 136164, 2020.

Zhang, N., Wu, Y., and Zhang, Q.: Forecasting the evolution of the sea ice in the Liaodong Bay using meteorological data, Cold Reg Sci Technol, 125, 21-30, 2016.

---

## Author Comment (AC3)

**Response to Reviewer #2**

We want to thank you for carefully reading and the constructive comments on our manuscript. Please find below our responses (in black) to all your comments (in blue) point by point, with new text added in the manuscript highlighted in *Italic*.

This paper describes the set up and validation of a coupled ocean-ice model over the Bohai Sea, with a focus on sea ice. The analysis was done using a 22-year hindcast simulation, and the intention is to use this model for long term and climate change studies - which a feature that makes it different to the other models developed for this area.

The presence of sea ice in the Bohai Sea, with it being at such a low latitude is interesting and it would be nice to include an explanation as to why it occurs.

Indeed, the Bohai Sea is the southernmost seasonal frozen sea in the Northern Hemisphere. The shallow water depth, low sea surface salinity, and geographical location are the main reasons for the formation of sea ice in the Bohai Sea. We have detailed this part to the introduction for clarity in the following.

*The formation of sea ice in the Bohai Sea mainly depends on the geographical environment and hydrometeorological characteristics (Ding, 1999). Specifically, the Bohai Sea is located in the continental shelf area, and the average water depth is only 18 m (Su and Wang, 2012), which indicates low oceanic heat content in winter. Following a northern continental climate, the Bohai Sea is affected by the cold Siberian air every winter, which causes the sea surface temperature of the Bohai Sea to be significantly lower than that at the same latitude (Zhang et al., 2016; Donlon et al., 2012). In addition, the sea surface salinity of the Bohai Sea is about 30 PSU, which is the lowest in the entire coastal waters in China (Yan et al., 2020). It means that the Bohai seawater freezes before reaching the maximum density. Therefore, it even more easily convects and loses heat just before freezing.*

Sea ice is the main focus of this paper however the underlying ocean dynamics of the Bohai Sea are not presented. I think that this article is lacking in the presentation of what is happening in the ocean and this needs to be addressed before publication.

We agree that the presentation of the underlying ocean is very valuable. We have added the comparisons of ocean temperature and salinity stratification between NEMO-Bohai and the atlas in a new section 3.2.4, as well as ocean circulation in section 3.2.3. In addition, more explanations for model biases are further proposed for clarify in section 3.

Specific points/questions:
1. You state that "a regional model for the Bohai Sea based on NEMO has not yet been developed, until now" but unfortunately this is not correct, as Li et al. 2021 have also set up a coupled NEMO model in this area:
Li, R., Lu, Y., Hu, X. et al. Space–time variations of sea ice in Bohai Sea in the winter of 2009–2010 simulated with a coupled ocean and ice model. J Oceanogr 77, 243–258 (2021). https://doi.org/10.1007/s10872-020-00566-2

Thanks for pointing out the study (Li et al., 2021), which simulated sea ice variations in the severe winter of 2009–2010 in the Bohai Sea based on NEMO 3.6 and distinguished significantly with our model application for climate studies. We have added this reference in the introduction part. At the same time, we have revised the mentioned sentence:

*However, a NEMO-based regional model for the Bohai Sea has not been attempted for long-term climate studies until now.*

2. When using FRS boundary conditions, the number of cells over which the relaxation is applied (nn_rimwidth) is typically between 8 and 10. I am interested as to why you have chosen to set it to 1 here.

This is a long story. We did not succeed in producing our boundary files by SIREN. Instead, we used the script provided by our coauthor, which is developed for NEMO application in the Barents and Kara Seas. In the script, the relaxation zone is set to be 1 grid cell, and the same setting can also be found in other studies (Thompson et al., 2021). Yes, the width of the FRS zone is typically set to a referenced value between 8 and 10. So far, we have only used this option (nn_rimwidth=1), but we will try some sensitivity experiments in the future.

3. It would be good to include some background information on the circulation of the ocean. Ocean currents play a role in determining the position of ice floes, so it would be beneficial to show that this model is simulating this properly before then going on to show how the model performs in predicting details in sea ice.

Thanks for your comments. We have added a new section 3.2.3, within which a new figure is added to illustrate the ocean currents at the surface and 16 m depth in August and February. We have also discussed the water volume exchange between the Bohai Sea and the Yellow Sea.

4. Instead of solely concentrating on surface plots, it would be good to include some vertical profiles of temperature compared to observations in the validation section. Perhaps also some transects of salinity and temperature across key areas in the results section, which would show the presence of any stratification of the ocean in different seasons.

Thanks for your suggestion, and we have added the figure and text in a new section (see section 3.2.4) accordingly to compare S/T vertical profiles:

*NEMO-Bohai and observed water temperature and salinity profiles along the transect AB (see Fig.1) are shown in Fig. 6 and Fig. 7, respectively. Observations are from the atlas by Chen (1992), which is based on data from the 1950s to 1990s. The temporally closest 5-year period from 1995 to 2000 of NEMO-Bohai simulations was selected for model-observation comparisons. Common features are found both in the model and observations. The Bohai Sea waters are vertically well-mixed in autumn and winter, and they have a remarkable homogeneous vertical distribution for both temperature and salinity. In spring and summer, thermal stratification occurs with a significant cold-water core at depth, eventually eroded in autumn. As apparent in Fig. 6 and Fig. 7, the stratification in shallow coastal waters is generally homogeneous. Similar features were reported by Wang et al. (2008), who analyzed the seasonal variations of the vertical profiles in the Bohai Sea.*

*The model results, however, show some discrepancies compared to the atlas. Although the model reproduces the summer saline stratification, it is weaker than in the atlas. Nonetheless, Li et al. (2015) reported that the summer salinity stratification in the Bohai Sea is possibly weaker than in the atlas, with an observed top-to-bottom salinity difference of 0.6 PSU. The modeled salinity distribution along with transect AB during summer is possibly affected by the high vertical diffusivity. In the north part of the transect, which corresponds to the northern Liaodong Bay, a negative salinity bias is visible compared to the atlas. In addition to the reasons mentioned in section 3.2.2, inaccuracies in the ETOPO1 bathymetry, especially in the low water depth region seen from Fig. 6 and Fig. 7, may also cause these underestimations.*

[Figure]

**Figure 6: Comparison of vertical profiles of water temperature (°C) along with transect AB (locations shown in figure 1) between NEMO-Bohai (a-d) and the atlas (Chen, 1992) (e-h) in February, May, August, and November.**

[Figure]

**Figure 7: Similar to figure 6 but for salinity (PSU).**

**Technical:**

\*There are quite a few mistakes in the language throughout the manuscript (too many to list here) and these need be corrected before publication.

We have followed you and checked grammar throughout the manuscript. We also have finished the English Language Editing service to improve English presentation accordingly.

\*On pages 6 – 10, the odd symbols made it quite hard to read whilst reviewing. This should also be corrected in the manuscript before publication.

Yes, there are some glitches on pages 6-10. The certain terms in Nimbus Mono L Font are garbled, and we have clarified in Author Comment (https://doi.org/10.5194/gmd-2021-100-AC1). We have revised these odd symbols in the revised manuscript.

\*Salinity units should be in PSU.

Changed.

\*Figure 4. Ensure that the number of ticks on the y axis is the same (c, f, h have more).

Thank you for noting this. The number of ticks in Fig. 4 has been unified for all subfigures.

\*Page 16, line 342: "Sea ice volume is defined as the total ice over the whole Bohai Sea, which is calculated through sea ice concentration times ice thickness in all grids."
This should be replaced by: "…sea ice concentration multiplied by ice thickness in all grids."

The sentence has been revised accordingly.

**Reference**

Chen, D. X.: Marine atlas of Bohai Sea, Yellow Sea and East China Sea: Hydrology, Ocean Press, Beijing, China, 1992.

Ding, D.W.: Introduction to sea ice engineering (in Chinese), Ocean Press, Beijing, China, 1999.

Donlon, C. J., Martin, M., Stark, J., Roberts-Jones, J., Fiedler, E., and Wimmer, W.: The operational sea surface temperature and sea ice analysis (OSTIA) system, Remote Sens Environ, 116: 140-158, 2012.

Li, R., Lu, Y., Hu, X., Guo, D., Zhao, P., Wang, N., Lee, K., and Zhang, B.: Space-time variations of sea ice in Bohai Sea in the winter of 2009–2010 simulated with a coupled ocean and ice model, J Oceanogr, 77(2), 243-258, 2021.

Li, Y., Wolanski, E., and Zhang, H.: What processes control the net currents through shallow straits? A review with application to the Bohai Strait, China, Estuar Coast Shelf S, 158, 1-11, 2015.

Su, H. and Wang, Y.: Using MODIS data to estimate sea ice thickness in the Bohai Sea (China) in the 2009-2010 winter, J Geophys Res-Oceans, 117, C10018, 2012.

Thompson, B., Sanchez, C., Heng, B. C. P., Kumar, R., Liu, J., Huang, X. Y., and Tkalich, P.: Development of a MetUM (v 11.1) and NEMO (v 3.6) coupled operational forecast model for the Maritime Continent–Part 1: Evaluation of ocean forecasts, Geosci Model Dev, 14(2), 1081-1100, 2021.

Wang, Q., Guo, X., and Takeoka, H.: Seasonal variations of the Yellow River plume in the Bohai Sea: A model study, J Geophys Res, 113, C08046, 2008.

Yan, Y., Uotila, P., Huang, K., and Gu, W.: Variability of sea ice area in the Bohai Sea from 1958 to 2015, Sci Total Environ, 709, 136164, 2020.

Zhang, N., Wu, Y., and Zhang, Q.: Forecasting the evolution of the sea ice in the Liaodong Bay using meteorological data, Cold Reg Sci Technol, 125, 21-30, 2016.

---

## Author Response (AR2)

Dear Editor and Reviewers:

Thank you very much for your thoughtful and constructive comments concerning our manuscript "NEMO-Bohai 1.0: a high-resolution ocean and sea ice modelling system for the Bohai Sea, China" (No. gmd-2021-100). The comments are valuable and helpful for improving our manuscript, which we have addressed in our point-by-point responses enclosed. In what follows we will bring the original comments in blue and the responses in black, with new text added in the manuscript highlighted in *Italic*. The changes made in the revised paper are highlighted in the track changes version of the manuscript attached.

Best regards,
Yu Yan and Petteri Uotila (on behalf of all the co-authors)

**Response to comments from Referee #1**

This is my second review of this manuscript, and I think there has been a great improvement from the first version. The language is a lot better and many additions have been done which give more backbone to the article.
I still have however a few points, mostly about ocean dynamics. I do not see these points as compulsory for publication, but I think they would improve the manuscript.

We appreciate the reviewer's recognition of our first revision, and thank for the second round of review on our manuscript. We have considered your comments to improve the manuscript and answered point by point.

- The authors mention that the circulation in the Bohai Sea is mostly barotropic. From Figure 5, I do not think it is that obvious, especially in the Northern part of the basin where the surface velocities go South whereas the deeper ones go North. Also the representation with vectors makes it very difficult to distinguish velocities close to the entrance of the Bohai Sea, I suggest you use stream functions instead. You could for example plot a barotropic stream function, but since there is a obviously a baroclinic circulation also plot one stream function for the upper layer and one for the lower layer, and compute the divergence of each layer which should correspond to the baroclinic circulation. I guess since the baroclinic circulation occurs in Winter and Summer then it is a circulation of haline nature, and because of the restricted size of the area it is obviously not completely geostrophic.

Yes, we agree that it is hard to distinguish velocities represented with original vectors plots in the Bohai Strait. Following your suggestions, we have used stream functions instead. In Fig.5, the black lines and arrows represent the streamlines and directions of the current vector field, respectively. Besides, as you indicated, the barotropic flow is not that obvious. Our statement on the barotropic circulation is a speculative one and thus we decided to delete this statement and study further in future study. We have also modified the corresponding text in section 3.2.3:

*The simulated monthly mean current velocities at the surface and 16 m depth in February and August are shown in Fig. 5. The monthly mean current velocities are calculated based on hourly model output during August 2012 and February 2013. The figure shows that both the sea surface*

*and 16 m depth currents are usually less than 0.4 m s⁻¹. Due to the blocking effects of the bays, the currents are weak at the head of the three bays, which is consistent with the observations by Chen et al. (1992). The maximum current velocity zone is located in the northern Bohai Strait, in a good agreement with the model simulation result of Ji et al. (2019). The inflow and outflow occur in the northern and southern parts of the Bohai Strait in both seasons, respectively, which is consistent with the observations (Zhang et al., 2018). Specifically, the strongest modeled inflow from the Yellow Sea through the Bohai Strait occurs in a narrow channel in its northern part, namely the Laotieshan Channel, which agrees with the observations of Wan et al. (2015). Also, Lin et al. (2011) suggested that persistent winds drive a cyclonic coastal current in the northern Yellow Sea, and one branch of the current enters the Bohai Sea at the northern Bohai Strait, which transports warm and saline water from the Yellow Sea.*

[Figure]

***Figure 5: Simulated monthly mean current velocities at surface and 16 m depth in August 2012 and February 2013. The monthly mean current velocities are calculated based on the outputs with hourly intervals. The black lines and arrows represent the streamlines and directions of the current vector field, respectively. The filled contours denote the current speed in m s⁻¹.***

- Not only there seem to be a baroclinic circulation, but also it is possible that it could be stronger in reality than in the model: the comparison between model and T/S profiles shows that there are thermal and haline frontal structures. Which brings me to a second point, the model is indeed too mixed, a fact the authors explain with bathymetry errors, and with which I do not agree. I think the

use of TKE with no specific tuning for such a region is responsible. Given the low inertia of the system, it is easy and it would be interesting to see how testing a turbulence closure that is more realistic would change the currents and perhaps even the sea ice cover. This can be easily done by switching to a GLS approach, or tuning the TKE scheme that you already use. For the latest, I suggest to refer to this article https://gmd.copernicus.org/articles/8/69/2015/, and more specifically to the Hp parameter, and the background diffusivity/viscosity.

Indeed, we agree that the model is too mixed. As for the difference between modeled and observed T/S profiles, especially for the lack of stratification in summer, it is possibly caused by the vertical mixing setting with the used TKE closure scheme, and the high setting of vertical diffusivity in the model. So far, our vertical mixing is actually strong, but we plan to carry out the sensitivity experiments with different turbulent closure schemes and vertical diffusivity coefficients. Following your suggestions, we have mentioned this in 3.2.4:

*The modeled salinity stratification in summer is weaker compared to the atlas, which is possibly caused by the vertical mixing setting with the used TKE closure scheme, and the high setting of vertical diffusivity in the model.*

Thanks for pointing out the study (Reffray et al., 2015), which provides a comprehensive overview of the turbulent vertical mixing options. We have added this reference in the perspective part (see section 5) and further emphasized the role of tuning vertical mixing:

*In addition, in order to carry out more accurate estimation of vertical mixing, it is worth implementing the experiments of turbulent vertical mixing options (Reffray et al., 2015) for the Bohai Sea for further development of NEMO-Bohai.*

Other remarks:
- Please check references, there are obviously bugs in some places. An example is Bernard (2006) which refers to Barnier (2006).

Thank you for noting this. We have followed you and checked the references. We have revised Bernard et al. (2006) to Barnier et al. (2006) accordingly.

*Barnier, B., Madec, G., Penduff, T., Molines, J. M., Treguier, A. M., Le Sommer, J., Beckmann, A., Biastoch, A., Böning, C., Dengg, J., Derval, C., Durand, E., Gulev, S., Remy, E., Talandier, C., Theetten, S., Maltrud, M., McClean, J., and De Cuevas, B.: Impact of partial steps and momentum advection schemes in a global ocean circulation model at eddy-permitting resolution, Ocean Dynam, 56(5), 543-567, 2006.*

- In table 1, please put a black separating column at the centre so that one does not read an entire line and falls into confusion.

Done.